# SE($n$)-Invariant Flow Matching: A General Framework with Application to Object Reassembly

**Gaël Heck** [1]  **Sylvie Le Hégarat-Mascle** [1]  **Nicolas Lermé** [1]

## Abstract

Reassembling $N$ fragments in $n$-dimensional space is a shape reconstruction task that is invariant to global rigid motions. Training directly on $\mathcal{M} = \mathrm{SE}(n)^N$ can be ill-posed: standard losses penalize solutions that differ only by a global transform. Existing methods often address this with ad-hoc anchoring which breaks permutation invariance across fragments and can introduce biases that must be mitigated with extensive and costly data augmentation. We propose a geometric framework that enforces invariance by construction. First, a **Global Gauge Fixing** (GGF) strategy deterministically aligns configurations using an intrinsic generalized-inertia rule. Second, we introduce a **quotient-invariant Flow Matching objective** that operates via orthogonal projection onto the horizontal tangent bundle. This construction factors out global pose at each timestep, enabling the model to learn only shape-changing dynamics on the quotient space $\mathcal{M}/\mathrm{SE}(n)$. Our unified $\mathrm{SE}(n)$-invariant framework admits efficient closed-form 2D/3D instantiations and improves accuracy on polygonal jigsaw puzzles and 3D fracture reassembly benchmarks.

## 1. Introduction

Reassembling fractured objects is an important and challenging inverse problem, spanning cultural-heritage restoration (e.g., RePAIR benchmark (Tsesmelis et al., 2024)), orthopedic fracture reduction (Fürnstahl et al., 2012), and geometric part assembly (Sellán et al., 2022). The breadth of applications, datasets, and solution paradigms is summarized in the recent survey (Lu et al., 2025). Given partial and noisy observations, the goal is to recover the relative poses of $N$ rigid fragments in $n$-dimensional space. A puzzle configuration is $x = (g_1, \dots, g_N) \in \mathcal{M} := \mathrm{SE}(n)^N$, yet the absolute pose of the reconstructed object is irrelevant: only its *shape* matters (Kendall, 1984; Littlejohn & Reinsch, 1997). Formally, configurations are equivalent along the *orbit* induced by the global $\mathrm{SE}(n)$ action, $[x] = \{h \cdot x : h \in \mathrm{SE}(n)\}$, i.e. the set of configurations obtained by applying the same rigid motion to the entire assembly. The intrinsic problem therefore lives on the quotient (shape) space $\mathcal{S} = \mathcal{M}/\mathrm{SE}(n)$. In principal-bundle terms, tangent directions decompose into *vertical* (pure global transport) and *horizontal* (intrinsic shape evolution) components (Kobayashi & Nomizu, 1996). Without handling degeneracy along $\mathrm{SE}(n)$ orbits, generative objectives can be dominated by arbitrary vertical drift, yielding ill-conditioned regression targets and slow convergence (Bronstein et al., 2021; Kvinge et al., 2022).

**From extrinsic heuristics to global geometry.** Most existing methods enforce invariance via anchoring (Wang et al., 2025a; Li et al., 2025), which breaks permutation symmetry (Zaheer et al., 2017) and introduces geometric bias. Anchoring can amplify errors for ambiguous references, while "best-anchor" heuristics introduce spurious correlations by freezing prominent parts; mitigating these biases often requires costly augmentation that only approximates the true symmetry. In 3D, additional surface cues (e.g., separating rough fractures from smooth exteriors) can reduce ambiguities (Li et al., 2025), but this can be brittle when such cues are weak, bringing back the ill-posedness of 2D assemblies (Li et al., 2025; Wang et al., 2025a).

We instead adopt a strictly intrinsic formulation. We define a canonical section $\Sigma \subset \mathcal{M}$ (centered configurations with aligned principal inertia axes) and rely on **Global Gauge Fixing**: a deterministic map $\Psi : \mathcal{M} \to \Sigma$ that selects a canonical representative per orbit (almost everywhere) and removes global degrees of freedom (Bullo & Lewis, 2005; Kobayashi & Nomizu, 1996). Since gauge fixing alone does not prevent drift, we propose a **quotient-invariant Flow Matching loss** that analytically projects residuals onto the horizontal tangent bundle using the inertia–momentum equations (Montgomery, 1993; O'Neill, 1966). The network thus learns only shape-changing dynamics, effectively operating on $\mathcal{S}$ rather than the full group (Cohen & Welling, 2016;

[1]SATIE Laboratory UMR 8029, Université Paris-Saclay, 4 avenue des sciences, Gif-sur-Yvette 91190, France. Correspondence to: Gaël Heck <gael.heck@universite-paris-saclay.fr>.

*Proceedings of the $43^{rd}$ International Conference on Machine Learning*, Seoul, South Korea. PMLR 306, 2026. Copyright 2026 by the author(s).

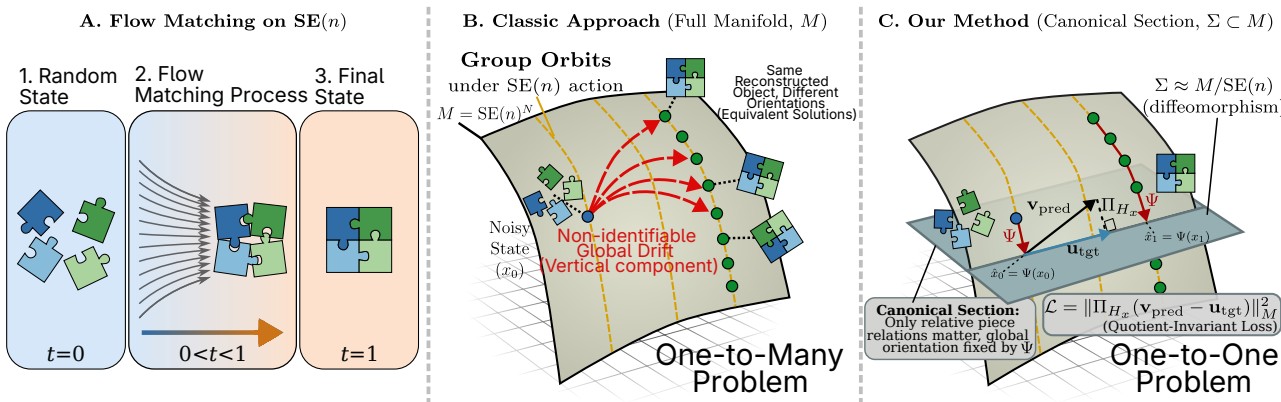

**A. Flow Matching on SE($n$)**

1. Random State
2. Flow Matching Process
3. Final State

$t=0$    $0<t<1$    $t=1$

**B. Classic Approach** (Full Manifold, $M$)

**Group Orbits**
under SE($n$) action
$M = SE(n)^N$

Same Reconstructed Object, Different Orientations (Equivalent Solutions)

Noisy State ($x_0$)

Non-identifiable Global Drift (Vertical component)

One-to-Many Problem

**C. Our Method** (Canonical Section, $\Sigma \subset M$)

$\Sigma \approx M/SE(n)$ (diffeomorphism)

$\mathbf{v}_{\text{pred}}$   $\Pi_{H_x}$   $\Psi$

$\mathbf{u}_{\text{tgt}}$   $\dot{\hat{x}}_1 = \Psi(x_1)$

$\hat{x}_0 = \Psi(x_0)$

**Canonical Section:** Only relative piece relations matter, global orientation fixed by $\Psi$

$\mathcal{L} = \|\Pi_{H_x}(\mathbf{v}_{\text{pred}} - \mathbf{u}_{\text{tgt}})\|_M^2$ (Quotient-Invariant Loss)

One-to-One Problem

*Figure 1.* **Intuitive overview.** (A) We reconstruct 2D or 3D objects by learning the flow from a disordered state to a reconstructed one. (B) Due to global symmetries, many configurations are equivalent, forming redundant "vertical" orbits. Learning on the full space is ambiguous because of unpredictable drift along these orbits. (C) We eliminate this redundancy by fixing a gauge: the mapping $\Psi$ projects configurations onto a unique slice (canonical section) across the orbits, while the operator $\Pi_{\mathcal{H}}$ projects the gradients (velocities) to remove irrelevant vertical drift. The model thus learns only meaningful "horizontal" motions that change the shape.

Bronstein et al., 2021).

**Contributions.** We combine this geometric framework with Flow Matching (Lipman et al., 2023) to learn an SE($n$)-invariant generative flow as summarized in Figure 1. Our contributions are threefold. (i) We **learn a SE($n$)-invariant vector field** by projecting the target velocity onto the horizontal bundle, ensuring the learned flow focuses purely on intrinsic shape changes. (ii) We establish **Global Gauge Fixing** as a robust deterministic canonicalization that selects one representative per orbit (on the regular set), reducing the problem dimension from $N \dim SE(n)$ to $(N-1) \dim SE(n)$. (iii) We design a **horizontally projected FM regression loss** from the analytic solution of the inertia-momentum equations, avoiding spurious gradients from the unidentifiable global gauge. The source code will be made available.

## 2. Related Work

**Generative models on geometric spaces.** Diffusion models (Song & Ermon, 2019; Ho et al., 2020) and their manifold extensions (De Bortoli et al., 2022; Song et al., 2023) have enabled high-quality generation beyond images, including structured geometric objects (e.g., proteins and poses) (Dhariwal & Nichol, 2021; Watson et al., 2023; Yim et al., 2023; Gong et al., 2023; Urain et al., 2023). Flow Matching (FM) (Lipman et al., 2023) provides a simulation-free alternative that regresses a time-dependent vector field transporting noise to data. It is particularly natural on SE(3) because it learns velocity fields; for rigid poses, velocities are twists in the Lie algebra $\mathfrak{se}(3)$ and can be integrated on the manifold without resorting to Euclidean parametrizations of rotation. FM has indeed been explored in domains including molecular structure and robotics (Bose et al., 2024;

Chisari et al., 2025). Quotient-Space Diffusion Models (QSDM) (Xu et al., 2026) also study quotient-space generative modeling, but in a diffusion setting for molecular and protein structure generation, where point-coordinate configurations in $\mathbb{R}^{3N}$ are considered up to global SE(3) transformations.

**2D puzzle solving and reassembly.** Classical jigsaw solvers relied on compatibility scores (Gallagher, 2012; Paikin & Tal, 2015) and graph optimization (Zhang & Li, 2014; Le & Li, 2019), with robust variants for cultural-heritage degradation (Derech et al., 2021) and geometric heuristics such as spring-mass systems (Harel & Ben-Shahar, 2021). Recent learning-based approaches introduce generative modeling of poses, notably coordinate diffusion for polygonal puzzles (Hossieni et al., 2023) and graph-diffusion formulations spanning 2D/3D assemblies (Scarpellini et al., 2024). ReAssembleNet (Islam et al., 2025) targets 2D fresco reconstruction using keypoint and appearance-driven diffusion.

**3D fracture reassembly.** Learning-based 3D reassembly progressed from Breaking Bad (Sellán et al., 2022) to matching and diffusion pipelines (Lu et al., 2023; Lee et al., 2024; Wang et al., 2025a; Scarpellini et al., 2024). GARF (Li et al., 2025) is a strong recent FM-based method for real-world fracture reassembly, combining fracture-aware representations with generative alignment on SE(3).

**Symmetry handling.** A central distinction across reassembly methods is how they handle the unidentifiable global SE(3) pose. Some approaches do not address this symmetry explicitly: they operate in absolute coordinates (Hossieni et al., 2023) or mitigate ambiguity empirically (e.g., via augmentation) without an explicit geo-

metric mechanism (Scarpellini et al., 2024). Beyond these pragmatic settings, methods that aim to treat the symmetry more directly typically (i) break it with an anchor/reference fragment (Islam et al., 2025; Wang et al., 2025a; Li et al., 2025), (ii) enforce equivariance by constraining the learned generative field or transport operator to respect group actions (Klein et al., 2023; Wang et al., 2025b), or (iii) use SE(3)-equivariant network architectures for discriminative pose prediction (Wu et al., 2023; Lim et al., 2025).

We summarize these symmetry-handling strategies in Tab. 1, which groups methods by mechanism (anchor/augmentation/equivariance/quotient), reports their output space and domain, and highlights the operating assumptions of each family.

# 3. Preliminaries

We recall the main notions underlying Flow Matching for continuous normalizing flows (CNFs), and the tractable conditional flow matching (CFM) objective used for learning the time-dependent vector field.

## 3.1. Continuous Normalizing Flows

A CNF (Chen et al., 2018; Grathwohl et al., 2019) defines a deterministic transport from a simple prior $p_0$ (typically $\mathcal{N}(0, I)$) to a target distribution $q$ through a time-dependent diffeomorphism $\phi_t : \mathbb{R}^d \to \mathbb{R}^d$ induced by the ODE

$$\frac{d\phi_t(x)}{dt} = v_t(\phi_t(x)), \qquad \phi_0(x) = x,$$

where $v_t$ is parameterized by a neural network $v_\theta(x, t)$. The induced probability path is $p_t = (\phi_t)_* p_0$ (where $(\phi_t)_*$ is the push-forward operator), and sampling amounts to drawing $x_0 \sim p_0$ and integrating the learned field from $t = 0$ to $t = 1$ to obtain $x_1 = \phi_1(x_0) \sim p_1 \approx q$.

## 3.2. Flow Matching and Conditional Flow Matching

Given a probability path $\{p_t\}_{t \in [0,1]}$ and a (generally unknown) vector field $u_t$ that transports $p_0$ along this path, Flow Matching (FM) trains $v_\theta$ by the regression objective (Lipman et al., 2023)

$$\mathcal{L}_{\text{FM}}(\theta) = \int_0^1 \mathbb{E}_{x \sim p_t} \left[ \|v_\theta(x, t) - u_t(x)\|_2^2 \right] dt,$$

which is typically intractable since neither $p_t$ nor $u_t$ are available in closed form. Conditional Flow Matching (CFM) circumvents this difficulty by introducing a family of conditional paths $p_t(x \mid x_1)$ for $x_1 \sim q$, chosen so that sampling $x \sim p_t(\cdot \mid x_1)$ and evaluating the associated conditional target field $u_t(x \mid x_1)$ are tractable (Lipman et al., 2023; Albergo & Vanden-Eijnden, 2023). The resulting objective

is

$$\mathcal{L}_{\text{CFM}}(\theta) = \mathbb{E}_{\substack{t \sim \mathcal{U}[0,1], \\ x_1 \sim q, \\ x \sim p_t(\cdot \mid x_1)}} \left[ |v_\theta(x, t) - u_t(x \mid x_1)|_2^2 \right].$$

and satisfies $\nabla_\theta \mathcal{L}_{\text{CFM}} = \nabla_\theta \mathcal{L}_{\text{FM}}$ (Lipman et al., 2023), enabling stable, simulation-free training via supervised regression.

## 3.3. Choice of Conditional Path

Different choices of $p_t(x \mid x_1)$ lead to distinct training dynamics. The CFM framework recovers diffusion-type Gaussian paths, such as the variance-preserving construction (Ho et al., 2020; Albergo & Vanden-Eijnden, 2023). Alternatively, optimal-transport-inspired interpolations yield straighter trajectories, often simplifying learning and numerical integration (Lipman et al., 2023). In particular, taking a Gaussian path with mean $(1 - t)x_0 + tx_1$ leads to the rectified-flow target field $u_t(x \mid x_1) = x_1 - x_0$ (Liu et al., 2023), which we adopt in this work.

## 3.4. Lie Group Background

The special orthogonal group $SO(n)$ consists of rotations $R \in \mathbb{R}^{n \times n}$ satisfying $R^\top R = I$ and $\det R = 1$. The special Euclidean group $SE(n)$ is the group of rigid motions $x \mapsto Rx + t$, with semidirect-product structure $SE(n) \cong \mathbb{R}^n \rtimes SO(n)$. For $N \in \mathbb{N}^*$, we denote by $SO(n)^N$ and $SE(n)^N$ the corresponding Cartesian products. The Lie algebra $\mathfrak{se}(n) = T_e SE(n)$ represents infinitesimal rigid motions (twists); tangent spaces are identified by left translation, i.e., $T_g SE(n) \simeq g\,\mathfrak{se}(n)$. The exponential map $\exp : \mathfrak{se}(n) \to SE(n)$ maps a twist to the associated finite motion, while the logarithm $\log : SE(n) \to \mathfrak{se}(n)$ provides local coordinates in the tangent space.

# 4. Method: SE($n$)-Invariant Flow Matching

**Informal overview.** Before introducing the formal quotient construction, we summarize the method in simple terms, following Fig. 1-C. We first place each candidate assembly in a canonical coordinate frame by centering the fragments and aligning their principal axes. We then remove, from the training signal, any motion that would translate or rotate all fragments together. The learned flow therefore focuses only on relative fragment motion, i.e., the motion that changes the assembled shape.

**Formally,** we propose an SE($n$)-invariant generative framework that learns assembly dynamics in relative pose space while remaining insensitive to arbitrary global rigid motion. Our method has three components. First, we use a simple velocity parameterization together with a quadratic metric on the tangent bundle $T(\text{SE}(n)^N)$ which enables

*Table 1.* **Symmetry-handling strategies under global** $\mathrm{SE}(n)$ **pose ambiguity.** Columns report the kind of method (Meth.), the way symmetry is handled (Sym. hdl), the output domain (Output sp.), whether the method targets $\mathrm{SE}(n)^N$ multi-body configurations (In-scope), and the main assumptions (Applicability assumptions).

| Reference | Meth. | Sym. hdl | Output sp. | In-scope | Applicability assumptions |
|---|---|---|---|---|---|
| (Hossieni et al., 2023) | Diff. | / | $\mathbb{R}^4$ | ✓ | 2D polygonal jigsaws; diffusion in a Euclidean pose parameterization (absolute coordinates) for fragment transformations. |
| (Scarpellini et al., 2024) | Diff. | Augment. | $\mathrm{SE}(2;3)^N$ | ✓ | Unified 2D/3D graph-diffusion over fragment poses (GNN + diff. denoising on piece graph). |
| (Islam et al., 2025) | Diff. | Anchor. | $\mathrm{SE}(2)^N$ | ✓ | 2D fresco reconstruction from images; diffusion over fragment placement driven by learned keypoints/appearance cues. |
| (Wang et al., 2025a) | Diff. | Anchor. | $\mathrm{SE}(3)^N$ | ✓ | 3D fracture assembly via diffusion denoising + verification/merging (agglomerative pipeline). |
| (Li et al., 2025) | FM | Anchor. | $\mathrm{SE}(3)^N$ | ✓ | 3D real-world fracture reassembly; fracture-aware pretraining (incl. fracture segmentation) + FM for 6-DoF alignment; handles missing/extraneous pieces. |
| (Lee et al., 2024) | Match. | Anchor. | $\mathrm{SE}(3)^N$ | ✓ | Deterministic multi-part 3D assembly via efficient point-cloud matching of mating surfaces (no generative sampling). |
| (Lu et al., 2023) | Match. | Anchor. | $\mathrm{SE}(3)^N$ | ✓ | 3D fracture assembly via fracture-surface segmentation + multi-part point matching + global alignment (Breaking Bad). |
| (Derech et al., 2021) | Match. | Anchor. | $\mathrm{SE}(2)^N$ | ✓ | Classical 2D archaeological puzzle solving from boundary compatibility under erosion/abrasion (non-learning / matching-based). |
| (Harel & Ben-Shahar, 2021) | Match. | Anchor. | $\mathrm{SE}(2)^N$ | ✗ | Specialized 2D "crossing-cuts" puzzles with strong geometric constraints (restricted puzzle family). |
| (Wang et al., 2025b) | FM | Group field | $\mathrm{SE}(3)^N$ | ✓ | 3D point-cloud part assembly; SE(3)-equivariant FM predicting multi-part relative poses (shape completion by alignment). |
| (Klein et al., 2023) | FM | Equiv. transp. | $S(N) \times O(D)$ | ✗ | Generic equivariant FM for particle/physics distributions (permutation/rotation symmetry), not rigid-body pose assembly. |
| (Lim et al., 2025) | FM | Lifting layer | $\mathrm{SE}(3)$ | ✗ | 6-DoF grasp synthesis for a single rigid object (N=1), conditioned on object geometry. |
| (Wu et al., 2023) | Reg. | Equiv. layers | $\mathrm{SE}(3)^N$ | ✓ | SE(3)-equivariant pose prediction for 3D part/shape assembly in a dataset-defined canonical frame (supervised regression). |
| Ours | FM | Quot. Inv. | $\mathcal{S}$, 2/3D | ✓ | Gauge-fixed configurations and horizontally projected velocities for invariant reassembly. |

closed-form least-squares removal of global rigid-body drift. Second, we apply **canonical gauge fixing** via a deterministic map $\Psi$ that selects a unique representative on a section $\Sigma$ for (almost) every $\mathrm{SE}(n)$-orbit. Beyond enforcing invariance, this canonicalization makes the Flow Matching (FM) supervision *single-valued* and *identifiable*, reducing target ambiguity and regression variance. Third, we define a **quotient-invariant objective** by projecting velocity residuals onto the horizontal (shape-changing) subspace, so the loss depends only on intrinsic assembly motion and not on global drift. We leverage the established geometric properties of shape spaces, referring to (Kendall, 1984) and (Kobayashi & Nomizu, 1996) for a rigorous treatment of gauge invariance and quotient manifolds.

Section 1 and Fig. 1 provide an intuitive account of motivation and the design of these components. For convenience, we restate the definition of $\Psi$ in Sec. 4.3 and give 2D/3D closed-form instantiations of the projected loss in Sec. 4.6.

### 4.1. Problem Formulation and Symmetries

We represent an $N$-part assembly by the rigid poses of its fragments $x = (g_1, \ldots, g_N) \in \mathcal{M} := G^N$ where $g_i = (\mathbf{p}_i, R_i) \in G := \mathrm{SE}(n)$. We aim to learn a generative flow on multi-body configurations that transports an initially random arrangement of fragments to the distribution of plausible assembled states.

We identify configurations under the diagonal left action $h \cdot$ $x = (hg_1, \ldots, hg_N)$ and work on the quotient $\mathcal{S} := \mathcal{M}/G$ (relative pose space). This matters for learning because supervision does not fix the global frame: many velocity fields on $\mathcal{M}$ induce the same evolution of relative geometry, differing only by an arbitrary rigid "whole-object drift". Direct FM regression on $T\mathcal{M}$ can therefore waste capacity fitting this unobservable component. The remainder of this section clarifies this point and introduces our strategy: we choose a consistent gauge and project velocities onto the shape-changing (frame-invariant) component.

### 4.2. Geometric Decomposition of the Tangent Space

We endow $\mathcal{M}$ with a simple quadratic inner product that treats each fragment equally. A tangent vector at $x = (g_i)_{i=1}^N$ can be written as $\dot{x} = (\dot{g}_i)_{i=1}^N \in T_x\mathcal{M}$, where $\dot{g}_i = (\dot{\mathbf{p}}_i, \dot{R}_i) \in T_{g_i}\mathrm{SE}(n)$. Using the standard identification $T_{(\mathbf{p},R)}\mathrm{SE}(n) \simeq \mathbb{R}^n \oplus \mathfrak{so}(n)$, we represent $\dot{g}_i$ by coordinates $(\dot{\mathbf{p}}_i, \boldsymbol{\omega}_i)$, where $\dot{\mathbf{p}}_i$ is the linear velocity and $\boldsymbol{\omega}_i$ the angular velocity, defined through

$$\dot{R}_i = \widehat{\boldsymbol{\omega}}_i R_i, \qquad \widehat{\boldsymbol{\omega}}_i \in \mathfrak{so}(n), \tag{1}$$

where $\widehat{(\cdot)}$ is the usual map from angular-velocity coordinates to a skew-symmetric matrix (e.g., $\widehat{\boldsymbol{\omega}}\,\mathbf{x} = \boldsymbol{\omega} \times \mathbf{x}$ in $n = 3$). We then endow $T_x\mathcal{M}$ with the product metric

$$\|\dot{x}\|_{\mathbf{M}}^2 = \sum_{i=1}^N \left( \|\dot{\mathbf{p}}_i\|^2 + \|\boldsymbol{\omega}_i\|^2 \right), \tag{2}$$

with $\mathbf{M} = \mathrm{diag}\big(I_n,\ I_{\dim SO(n)}\big)$. This metric will be used to define an orthogonal projection that removes global rigid-body drift. At any point $x \in \mathcal{M}$, the tangent space splits as

$$T_x\mathcal{M} = \mathcal{V}_x \oplus \mathcal{H}_x. \tag{3}$$

**Vertical space $\mathcal{V}_x$.**  The vertical space is the tangent space to the diagonal $SE(n)$-orbit through $x$: it consists of velocities obtained by applying the *same* infinitesimal rigid motion $\eta = (\mathbf{v}, \boldsymbol{\omega}) \in \mathfrak{se}(n)$ to *each* fragment. Formally, we define the infinitesimal action $\alpha_x : \mathfrak{se}(n) \to T_x\mathcal{M}$ by

$$\forall i \in [\![1, N]\!], \quad \alpha_x(\eta)_i \;=\; \begin{bmatrix} \mathbf{v} + \widehat{\boldsymbol{\omega}}\,\mathbf{p}_i \\ \boldsymbol{\omega} \end{bmatrix}, \tag{4}$$

and set $\mathcal{V}_x = \mathrm{Im}(\alpha_x)$.

**The Horizontal Space $\mathcal{H}_x$.**  We define $\mathcal{H}_x := \mathcal{V}_x^{\perp}$ as the orthogonal complement with respect to the inner product induced by $\mathbf{M}$. Velocities in $\mathcal{H}_x$ represent pure shape evolution: they change relative geometry while carrying no global rigid-body drift under the chosen metric.

### 4.3. Global Gauge Fixing

To remove global pose redundancy, we define a canonical section $\Sigma \subset \mathcal{M}$ that intersects each $SE(n)$-orbit (almost everywhere) exactly once. We construct a deterministic gauge map $\Psi : \mathcal{M} \to \Sigma$ by centering and aligning principal axes of the centered point cloud. Given $x = (\mathbf{p}_i, R_i)_{i=1}^N$, we center positions by $\bar{\mathbf{p}} = \frac{1}{N}\sum_i \mathbf{p}_i$ and $\mathbf{p}_i^c = \mathbf{p}_i - \bar{\mathbf{p}}$. We then form the empirical covariance $C = \frac{1}{N}\sum_i \mathbf{p}_i^c(\mathbf{p}_i^c)^{\top}$, compute an eigendecomposition $C = U\Lambda U^{\top}$, and set $R^* = U^{\top}$ using a deterministic sign convention to enforce $\det(R^*) = 1$. The canonicalized configuration $\Psi(x) = (\tilde{\mathbf{p}}_i, \tilde{R}_i)_{i=1}^N = \tilde{x}$ is

$$\tilde{\mathbf{p}}_i = R^*\mathbf{p}_i^c, \qquad \tilde{R}_i = R^*R_i. \tag{5}$$

The gauge is unique on the regular set where $\Lambda$ has distinct eigenvalues. App. A.2 describes a deterministic tie-breaking convention for the remaining cases, yielding a well-defined map $\Psi$. In our experiments, smoother optimization-based canonicalizations (e.g., Riemannian Fréchet means on $SE(n)$) offered no benefit and were consistently, albeit slightly, outperformed by PCA despite their higher cost; see App. A.2 for an ablation.

### 4.4. Geodesic Probability Path

We define the probability path $p_t(x)$ via **geodesic interpolation** on the manifold $\mathcal{M}$. Given a data sample $x_1 \sim q$ and a noise sample $x_0 \sim p_0$, we first canonicalize both endpoints: $\tilde{x}_0 = \Psi(x_0)$ and $\tilde{x}_1 = \Psi(x_1)$.

The interpolant $x_t$ is defined component-wise. For each fragment $i \in [\![1, N]\!]$, the pose $g_i(t) = (\mathbf{p}_i(t), R_i(t))$ evolves as:

$$\begin{aligned} \mathbf{p}_i(t) &= (1-t)\tilde{\mathbf{p}}_{0,i} + t\tilde{\mathbf{p}}_{1,i}, \\ R_i(t) &= \exp\Big(t\,\log\big(\tilde{R}_{1,i}\tilde{R}_{0,i}^{\top}\big)\Big)\,\tilde{R}_{0,i}. \end{aligned} \tag{6}$$

By construction, $\mathbf{p}_i(0) = \tilde{\mathbf{p}}_{0,i}$, $\mathbf{p}_i(1) = \tilde{\mathbf{p}}_{1,i}$, and $R_i(0) = \tilde{R}_{0,i}$, $R_i(1) = \tilde{R}_{1,i}$.

The target velocity field $u_t(x_t)$ is defined as the tangent to this interpolant. In our velocity coordinates it is constant along the path and given by

$$u_i^{\mathrm{tgt}} \;=\; \begin{bmatrix} \tilde{\mathbf{p}}_{1,i} - \tilde{\mathbf{p}}_{0,i} \\ \log\big(\tilde{R}_{1,i}^{\top}\tilde{R}_{0,i}\big) \end{bmatrix}. \tag{7}$$

The rotational component is expressed in Lie-algebra coordinates: $\widehat{\omega}_i = \log(\tilde{R}_{0,i}^{\top}\tilde{R}_{1,i})$ while $\dot{R}_i(t) = \widehat{\omega}_i R_i(t)$.

### 4.5. Quotient-Invariant Flow Matching

We define a **quotient-invariant FM** loss by removing, at each $x_t$, the best-fitting global rigid-body drift and measuring the residual only in the horizontal subspace $\mathcal{H}_{x_t}$.

Let $\delta = v_\theta(x_t, t) - u_t(x_t) \in T_{x_t}\mathcal{M}$ be the raw residual, written in our velocity coordinates as $\delta_i = (\delta_{\mathbf{v},i}, \delta_{\boldsymbol{\omega},i})$. We obtain the horizontal residual by subtracting the $\mathbf{M}$-orthogonal projection of $\delta$ onto the vertical space $\mathcal{V}_{x_t}$:

$$\begin{aligned} \Pi_{\mathcal{H}_{x_t}}(\delta) &= \delta - \alpha_{x_t}(\eta^*), \\ \eta^* &= \arg\min_{\eta \in \mathfrak{se}(n)} \|\delta - \alpha_{x_t}(\eta)\|_{\mathbf{M}}^2. \end{aligned} \tag{8}$$

Equivalently, $\eta^*$ is the solution of $\alpha_{x_t}^{\top}\mathbf{M}\,\alpha_{x_t}\,\eta = \alpha_{x_t}^{\top}\mathbf{M}\,\delta$,

$$\begin{aligned} \eta^* &= (\alpha_{x_t}^{\top}\mathbf{M}\,\alpha_{x_t})^{-1}\alpha_{x_t}^{\top}\mathbf{M}\,\delta = \mathcal{I}(x_t)^{-1}\mathcal{P}(x_t), \\ \mathcal{I}(x) &= \alpha_x^{\top}\mathbf{M}\,\alpha_x, \qquad \mathcal{P}(x) = \alpha_x^{\top}\mathbf{M}\,\delta. \end{aligned} \tag{9}$$

For $SE(2)$ and $SE(3)$, $\eta^*$ is given by a small weighted least-squares solve. In the centered gauge, this solve simplifies leading to the explicit formulas in Sec. 4.6.

Finally, the training objective is the expected energy of the projected horizontal residual:

$$\mathcal{L}(\theta) = \mathbb{E}_{t,x_0,x_1}\left[\|\Pi_{\mathcal{H}_{x_t}}(\delta)\|_{\mathbf{M}}^2\right]. \tag{10}$$

### 4.6. Practical Instantiations

In the canonical gauge, positions are centered (i.e., $\sum_i \mathbf{p}_i = \mathbf{0}$): the best global translational drift is the average residual, $\bar{\delta}_{\mathbf{v}} = \frac{1}{N}\sum_{i=1}^N \delta_{\mathbf{v},i}$, so we work with centered translational residuals $\delta_{\mathbf{v},i}^c = \delta_{\mathbf{v},i} - \bar{\delta}_{\mathbf{v}}$ (equivalently, $\sum_i \delta_{\mathbf{v},i}^c = 0$). After removing translation, the only remaining global degree of freedom is a *common* angular velocity $\boldsymbol{\omega}^*$. We obtain $\boldsymbol{\omega}^*$ by a $\mathbf{M}$-weighted least-squares fit of a single rigid-body rotation to the centered residuals, which yields the simple expressions below for $SE(2)$ and $SE(3)$, where $\mathbf{q}_i = \mathbf{p}_i^c$.

**Planar case (SE(2)).** Let $\delta_{\omega,i} \in \mathbb{R}$ denote the scalar rotational residual. Define

$$I_{\text{rot}} = \sum_{i=1}^{N} \|\mathbf{q}_i\|^2 + N, \qquad \tau = \sum_{i=1}^{N} (\mathbf{q}_i \times \delta_{\mathbf{v},i}^c) + \sum_{i=1}^{N} \delta_{\omega,i}. \tag{11}$$

where $\mathbf{a} \times \mathbf{b} := a_1 b_2 - a_2 b_1$. The optimal global angular velocity is $\omega^* = \tau / I_{\text{rot}}$, and the horizontal projection is

$$\Pi_{\mathcal{H}}(\delta)_i = \begin{bmatrix} \delta_{\mathbf{v},i}^c - \omega^* J \mathbf{q}_i \\ \delta_{\omega,i} - \omega^* \end{bmatrix}, \qquad J = \begin{bmatrix} 0 & -1 \\ 1 & 0 \end{bmatrix}. \tag{12}$$

**Spatial case (SE(3)).** With $\delta_{\boldsymbol{\omega},i} \in \mathbb{R}^3$ the rotational residual, the inertia matrix and angular momentum are:

$$\begin{aligned} \mathbf{J} &= \sum_{i=1}^{N} \left( \|\mathbf{q}_i\|^2 \mathbf{I} - \mathbf{q}_i \mathbf{q}_i^\top \right) + N\,\mathbf{I}, \\ \mathbf{L} &= \sum_{i=1}^{N} \left( \mathbf{q}_i \times \delta_{\mathbf{v},i}^c + \delta_{\boldsymbol{\omega},i} \right). \end{aligned} \tag{13}$$

The optimal global angular velocity is $\boldsymbol{\omega}^* = \mathbf{J}^{-1}\mathbf{L}$, and the projected residuals are

$$\Pi_{\mathcal{H}}(\delta)_i = \begin{bmatrix} \delta_{\mathbf{v},i}^c - \boldsymbol{\omega}^* \times \mathbf{q}_i \\ \delta_{\boldsymbol{\omega},i} - \boldsymbol{\omega}^* \end{bmatrix}. \tag{14}$$

These closed-form projections are differentiable and implemented efficiently as batched tensor operations.

**Inference Dynamics.** Sampling consists of integrating the learned vector field starting from a canonicalized noise sample $\tilde{x}_0 = \Psi(x_0)$. To ensure the trajectory evolves on the quotient (i.e., up to global rigid motion), we project the predicted velocity onto the horizontal space at every integration step:

$$\frac{dx}{dt} = \Pi_{\mathcal{H}_x}\big(v_\theta(x,t)\big). \tag{15}$$

In our velocity coordinates, writing $\Pi_{\mathcal{H}_x}(v_\theta(x,t)))_i = (\dot{\mathbf{p}}_i, \boldsymbol{\omega}_i)$, this means $\dot{\mathbf{p}}_i$ for translations and $\dot{R}_i = \widehat{\boldsymbol{\omega}}_i R_i$ for rotations. This projection removes the instantaneous vertical component, so generation depends only on shape change and is invariant to the choice of global frame. We refer to App. D for training and inference algorithms.

## 5. Experiments

We evaluate our framework on two settings: 2D polygonal puzzle solving and 3D fractured object reassembly. We first describe the architecture and the experimental setup, then ablation studies validating our geometric components, and finally we compare to state-of-the-art (SOTA) methods.

### 5.1. Experimental Setup

**2D simulated benchmarks.** We evaluate on **SVJP** (Square Voronoi Jigsaw Puzzle) (Hossieni et al., 2023), a synthetic dataset of $100,000$ square-domain Voronoi tessellations. Following (Hossieni et al., 2023), vertex coordinates are perturbed with i.i.d. Gaussian noise at $0\%$, $0.5\%$ and $1\%$ of the square width. We also evaluate on **SVFP**, derived from SVJP by replacing straight edges with irregular boundaries and introducing material degradation via stochastic erosion (gap levels $0.5\%$ and $1\%$). We report *Overlap* (area alignment between predicted and ground truth positions) and adjacency *Precision/Recall* computed with a Chamfer distance threshold $\tau = 1\%$ of the square's width.

**Floorplans.** We evaluate floorplan reconstruction on **MagicPlan** (98,780 houses) and **RPLAN** (60,000 floorplans), reporting *Mean Positional Error (MPE)* and *Graph Edit Distance (GED)*. MPE is the average pixel distance between predicted and ground-truth room centers on a $256 \times 256$ canvas. For GED, we build the adjacency graph by connecting rooms whose door centers are within $5$ pixels (following (Hossieni et al., 2023)).

**Archaeological reassembly.** We evaluate archaeological reassembly on **RePAIR** (Tsesmelis et al., 2024), which contains 121 frescoes with irregular fragment geometries, eroded, and missing pieces. We report translation RMSE ($\mathcal{T}_{mm}$, mm), rotation RMSE ($\mathcal{R}^\circ$, degrees), and *Overlap*.

**2D baselines.** On SVJP/SVFP, we compare against geometric generative baselines: score-based diffusion (Song et al., 2021) and FM (Lipman et al., 2023), instantiated respectively in Euclidean space $\mathbb{R}^3$ and in the manifold SE(2) (De Bortoli et al., 2022; Chen & Lipman, 2024). Our results are also compared against PuzzleFusion (Hossieni et al., 2023) on SVJP, SVFP, RPLAN and MagicPlan. On RePAIR (Tsesmelis et al., 2024), we report results against prior methods (Islam et al., 2025; Huang et al., 2006; Derech et al., 2021). In the tables, baseline results are obtained by running the authors' released models/code for inference under our evaluation protocol; *italicized* entries are quoted from the original papers.

**3D benchmarks.** We follow the GARF evaluation protocol (Li et al., 2025) and report $\text{RMSE}(R)$ and $\text{RMSE}(T)$, *Part Accuracy* (PA fraction of parts with Chamfer distance $< 10^{-2}$), and global Chamfer Distance (CD). We evaluate on the volume-constrained version of the **Breaking Bad** dataset (Sellán et al., 2022) (Everyday and Artifact subsets) and the **FRACTURA** dataset (real and synthetic fractures).

**3D baselines.** We compare against graph-based methods (Scarpellini et al., 2024), architecture-based equivariant approaches (Wu et al., 2023) and SOTA assembly pipelines including Jigsaw (Lu et al., 2023), PMTR (Lee et al., 2024), PuzzleFusion++ (Wang et al., 2025a), (Wang et al., 2025b),

*Table 2.* Effect of (i) canonical gauge fixing, (ii) the quotient-invariant loss, and (iii) multi-anchor supervision. We also report a test-time ablation with and without prediction projection on the SVJP dataset. $O$, $P$ and $R$ stand for **Overlap**, **Precision** and **Recall**, respectively.

| | Gauge fix. | Quot. loss | Anchor | O↑ | P↑ | R↑ |
|---|---|---|---|---|---|---|
| (A) Base | ✗ | ✗ | ✗ | .80 | .80 | .75 |
| (B) | ✓ | ✗ | ✗ | .90 | .89 | .88 |
| (B') | ✗ | ✓ | ✗ | .92 | .89 | .89 |
| (C) | ✓ | ✓ | ✗ | .93 | .90 | .89 |
| (D) Multi-anchor | ✗ | ✗ | ✓ | .87 | .85 | .83 |
| Test-time ablation | Prediction projection | | | | | |
| (C) w/o proj. (test) | ✗ | | | .93 | .90 | .89 |
| (C') with proj. (test) | ✓ | | | .94 | .91 | .90 |

*Table 3.* Effect of (i) canonical gauge fixing, (ii) the quotient-invariant loss, and (iii) test-time prediction projection on the Everyday and Artifact subsets of the Breaking Bad dataset. RMSE($R$), RMSE($T$) and CD are lower-is-better metrics, while PA is higher-is-better.

| | Components | | | Metrics | | | |
|---|---|---|---|---|---|---|---|
| Variant | Gauge fix. | Quot. loss | Proj. | RMSE($R$)↓ | RMSE($T$)↓ | PA↑ | CD↓ |
| **Everyday** | | | | | | | |
| (A) Base | ✗ | ✗ | ✗ | 9.67 | 3.23 | 91.48 | 0.76 |
| (B) | ✓ | ✗ | ✗ | 6.98 | 1.54 | 94.76 | 0.26 |
| (B') | ✗ | ✓ | ✗ | 6.07 | 1.29 | 95.34 | 0.22 |
| (C) | ✓ | ✓ | ✗ | 5.74 | 1.13 | 95.54 | 0.19 |
| (C') with proj. (test) | ✓ | ✓ | ✓ | 5.80 | 1.12 | 95.91 | 0.19 |
| **Artifact** | | | | | | | |
| (A) Base | ✗ | ✗ | ✗ | 10.02 | 3.98 | 90.21 | 0.98 |
| (B) | ✓ | ✗ | ✗ | 6.02 | 1.36 | 94.93 | 0.41 |
| (B') | ✗ | ✓ | ✗ | 5.76 | 1.30 | 95.01 | 0.37 |
| (C) | ✓ | ✓ | ✗ | 5.41 | 1.12 | 95.23 | 0.34 |
| (C') with proj. (test) | ✓ | ✓ | ✓ | 5.36 | 1.01 | 95.72 | 0.33 |

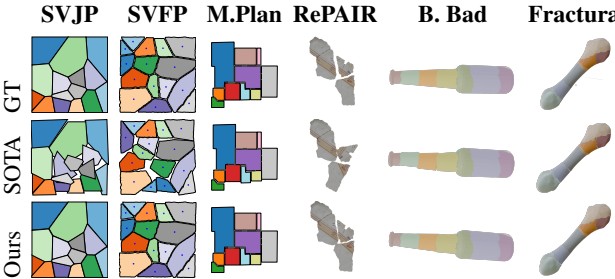

*Figure 2.* Qualitative reconstruction results on six benchmarks. Rows show Ground Truth (GT), the strongest baseline, and our method. Columns correspond to SVJP (Hossieni et al., 2023), SVFP, MagicPlan (Hossieni et al., 2023), RePAIR (Tsesmelis et al., 2024), Breaking Bad (Sellán et al., 2022), and FRACTURA (Li et al., 2025).

QSDM (Xu et al., 2026) and GARF (Li et al., 2025). For the latter, two settings were considered: While the GARF fracture encoder is trained on a 1.9M fragment dataset, we also evaluate GARF-mini, which shares the exact same architecture but is pre-trained exclusively on the Everyday subset of Breaking Bad. Like for 2D experiments, in all tables, non-reproduced entries are shown in *italics*.

*Table 4.* Quantitative results on 2D benchmarks. SVJP is evaluated with three vertex-noise levels (0, 0.5%, 1.0%) (Hossieni et al., 2023). Cross marks indicate not applicable. Best values are in **bold**; second-best values are underlined. Reported values are averaged over ten runs.

| | Overlap↑ | | | Precision↑ | | | Recall↑ | | |
|---|---|---|---|---|---|---|---|---|---|
| Noise Level | 0 | .5% | 1% | 0 | .5% | 1% | 0 | .5% | 1% |
| **SVJP** | | | | | | | | | |
| (Song et al., 2021) | .66 | .63 | .58 | .54 | .49 | .43 | .57 | .53 | .49 |
| (Lipman et al., 2023) | .65 | .65 | .56 | .55 | .51 | .42 | .58 | .52 | .47 |
| (De Bortoli et al., 2022) | .79 | .78 | .69 | .73 | .70 | .62 | .70 | .67 | .60 |
| (Chen & Lipman, 2024) | .80 | .80 | .68 | .74 | .71 | .64 | .69 | .65 | .60 |
| (Hossieni et al., 2023) | .76 | .71 | .60 | .80 | .72 | .60 | .74 | .64 | .44 |
| **Ours** | **.93** | **.89** | **.78** | **.91** | **.87** | **.74** | **.88** | **.84** | **.71** |
| **SVFP** | | | | | | | | | |
| (Hossieni et al., 2023) | ✗ | ✗ | ✗ | ✗ | ✗ | ✗ | ✗ | ✗ | ✗ |
| (Song et al., 2021) | .73 | .67 | .64 | .61 | .59 | .48 | .64 | .53 | .56 |
| (Lipman et al., 2023) | .72 | .70 | .62 | .60 | .55 | .49 | .63 | .57 | .54 |
| (De Bortoli et al., 2022) | .80 | .79 | .72 | .75 | .72 | .66 | .70 | .68 | .61 |
| (Chen & Lipman, 2024) | .83 | .84 | .76 | .76 | .74 | .66 | .71 | .66 | .61 |
| **Ours** | **.94** | **.87** | **.83** | **.90** | **.85** | **.83** | **.88** | **.86** | **.82** |

## 5.2. Architecture and Implementation

Our FM formulation is easy to plug into existing SOTA pipelines without modifying their feature extractors. In our experiments, we therefore instantiate it on top of published backbones whenever available.

**2D backbone.** To our knowledge, no standard backbone exists for reassembling arbitrary geometry-only (apictorial) polygons, we propose task-specific shape encoders to represent individual fragments. These embeddings are subsequently processed by a conventional attention-based encoder, following established baselines (see App. B for details). For the RePAIR dataset, we adopt the SOTA backbone (Islam et al., 2025) to leverage available textural cues.

**3D backbone.** We adopt the point-cloud backbone of GARF (Li et al., 2025) to isolate the contribution of our FM formulation from architectural changes. We disable all anchoring features during both training and sampling, including anchor-based data augmentation.

**Training & inference.** Experiments were run on NVIDIA H100 80GB GPUs using Python 3.12 and PyTorch 2.7.1. The number of GPUs, epochs, and batch size vary by settings and are reported in App. C.3. All models were trained with AdamW. For sampling from noisy state to predicted reconstruction, we integrate the learned vector field with Heun's method with 50 steps. Unless otherwise stated, quantitative results obtained from our runs are averaged over ten runs. Implementation details and per-benchmark hyperparameters are provided in App. C.

*Table 5.* Comparison with (Hossieni et al., 2023) on RPLAN and MagicPlan data. Best values are in **bold**; second-best values are underlined. Reported values are averaged over ten runs.

| Dataset | RPLAN | | MagicPlan | |
|---|---|---|---|---|
| Metric | MPE (↓) | GED (↓) | MPE (↓) | GED (↓) |
| (Hossieni et al., 2023) (Piece-level attention) | *46.18* | *2.27* | *53.11* | *6.41* |
| (Hossieni et al., 2023) (Vertex-level attention) | 11.69 | 0.99 | 39.28 | 3.03 |
| **Ours** | **7.20** | **0.71** | **30.11** | **2.50** |

*Table 6.* Comparison to SOTA methods for *archaeological* fragment reassembly on the RePAIR dataset (Tsesmelis et al., 2024). Best values are in **bold**; second-best values are underlined. Reported values are averaged over ten runs.

| Method | $Overlap \uparrow$ | RMSE ($\mathcal{R}°$) ↓ | RMSE ($\mathcal{T}_{mm}$) ↓ |
|---|---|---|---|
| (Derech et al., 2021) | *0.04* | *80.96* | *139.49* |
| (Tsesmelis et al., 2024) (Genetic Opti) | *0.05* | *85.63* | *151.71* |
| (Tsesmelis et al., 2024) (Greedy) | *0.02* | *76.99* | *135.95* |
| (Scarpellini et al., 2024) | *0.10* | *123.42* | *280.76* |
| (Zhou et al., 2024) | *0.13* | *91.54* | *364.62* |
| (Islam et al., 2025) | 0.21 | 29.20 | 16.01 |
| **Ours** | **0.41** | **21.83** | **10.72** |

## 5.3. Ablations

Table 2 isolates the impact of enforcing global $SE(2)$ invariance. We focus on SVJP to stress geometric ambiguity and to measure improvements attributable to the reconstruction process rather than local appearance cues. Canonical gauge fixing yields the largest gain (row B vs. row A: $+0.10$ Overlap), indicating that removing global-frame ambiguity substantially simplifies learning. The quotient-invariant loss further improves performance (row C vs. row B). At test-time, we additionally ablate prediction projection by evaluating the same trained model (row C) with vs. without projection (row C vs. row C'), which provides a consistent boost. Multi-anchor supervision improves over the unconstrained baseline (row D vs. row A) but remains below gauge fixing (row B), suggesting that symmetry breaking through anchors is less effective than invariance by construction. We refer the reader to the App. A for details about our ablation results and for experiments with alternative gauge choice beyond the PCA gauge.

Table 3 presents a directly comparable ablation in 3D settings (Breaking Bad dataset) and illustrates similar behavior.

## 5.4. Comparisons

Figure 2 provides qualitative reconstructions across all benchmarks, illustrating typical failure modes and the improved global consistency of our approach.

**SVJP.** Table 4 reports Overlap/Precision/Recall averaged over ten runs at three vertex-noise levels, while training is performed on clean shapes. On the in-distribution setting (0 noise), our method reaches 0.93 Overlap, compared to 0.80 for the strongest $SE(2)$ baseline (Chen & Lipman, 2024). As noise increases, performance drops for all methods, but our method remains consistently best across all noise levels,

including $1\%$ noise (Overlap 0.78 vs. 0.69 for the strongest baseline).

**SVFP.** On SVFP, enforcing $SE(2)$ invariance substantially improves robustness under degradation. At $1\%$ noise, our method improves Overlap from 0.76 to 0.83, Precision from 0.66 to 0.83 and Recall from 0.61 to 0.82 over the strongest $SE(2)$ baseline. Overall, invariance by construction reduces failure modes under distribution shift and yields reconstructions that remain both accurate and more complete.

**RPLAN and MagicPlan.** Table 5 reports floorplan reconstruction results on RPLAN and MagicPlan. Vertex-level attention (Hossieni et al., 2023) attends over all contour vertices (quadratic in total vertices), while piece-level attention compresses each fragment into a token and attends over fragments. Our method achieves the lowest error on both datasets, improving over the best PuzzleFusion variant (vertex-level attention): on RPLAN, MPE (lower is better) decreases from 11.69 to 7.20 and GED (lower is better) from 0.99 to 0.71; on MagicPlan, MPE decreases from 39.28 to 30.11 and GED from 3.03 to 2.50.

**RePAIR (archaeological reassembly).** Table 6 compares to prior methods on RePAIR (Tsesmelis et al., 2024). Our approach substantially improves all metrics, achieving 0.41 Overlap versus 0.21 for the strongest baseline (Islam et al., 2025), while also reducing rotation RMSE from $29.20°$ to $21.83°$ and translation RMSE from 16.01 mm to 10.72 mm. These gains indicate more accurate and more complete reconstructions on real archaeological fragments.

**3D benchmark (Breaking Bad and FRACTURA)** Table 7 reports 3D quantitative results under the GARF protocol on the Breaking Bad (Everyday/Artifact) and the FRACTURA datasets. Prior to our work, GARF (Li et al., 2025) is a strong recent baseline under this evaluation protocol; we therefore *re-run* GARF using the authors' released code and report the verified numbers here. We compare in two matched training regimes (Mini and full), and the colored deltas indicate the difference to the corresponding GARF baseline in the same regime (Ours (Mini) vs. GARF (Mini), and Ours vs. GARF). Across all three test sets, our $SE(3)$-invariant formulation consistently improves over GARF, reducing both rotation and translation errors while increasing Part Accuracy (PA) and lowering Chamfer Distance (CD). In particular, on Breaking Bad (Everyday), we improve $\mathrm{RMSE}(R)$ from 6.43 to 5.80 and $\mathrm{RMSE}(T)$ from 1.26 to 1.12, while increasing PA from $95.33\%$ to $95.91\%$ and reducing CD from 0.22 to 0.19. Similar gains hold on the Artifact subset and on FRACTURA ($\mathrm{RMSE}(R)$ $17.98 \rightarrow 17.20$, $\mathrm{RMSE}(T)$ $4.50 \rightarrow 4.03$, PA $84.18\% \rightarrow 85.13\%$, CD $6.36 \rightarrow 5.29$).

*Table 7.* 3D quantitative results under the GARF protocol on Volume-Constrained Breaking Bad Everyday/Artifact and FRACTURA (Li et al., 2025). We report $\text{RMSE}(R^\circ)$ (deg)↓, $\text{RMSE}(T) \downarrow (10^{-2})$, part accuracy (PA)↑, and Chamfer distance (CD)↓ $(10^{-3})$. Colored deltas are relative to the corresponding GARF setting (Ours (Mini) vs. GARF (Mini), Ours vs. GARF). Best values are in **bold**; second-best values are underlined. Reported values are averaged over ten runs.

| Methods | $\text{RMSE}(R^\circ) \downarrow$ | $\text{RMSE}(T) \downarrow$ | PA ↑ | CD ↓ |
|---|---|---|---|---|
| **Breaking Bad (Everyday) subset** | | | | |
| (Wu et al., 2023) | 79.2 | 15.1 | 26.98 | 12.33 |
| (Scarpellini et al., 2024) | *73.30* | *14.80* | *27.50* | - |
| (Lu et al., 2023) | 43.72 | 7.02 | 68.29 | 8.97 |
| (Lee et al., 2024) | *31.57* | *9.95* | *70.60* | *5.56* |
| (Wang et al., 2025b) | 81.21 | 16.13 | 26.93 | 12.49 |
| (Wang et al., 2025a) | 35.20 | 6.16 | 76.50 | 2.58 |
| (Xu et al., 2026) | 7.28 | 1.54 | 93.28 | 0.43 |
| (Li et al., 2025) (Mini) | 6.99 | 1.38 | 94.30 | 0.26 |
| (Li et al., 2025) | 6.43 | 1.26 | 95.15 | 0.22 |
| **Ours (Mini)** | $6.21^{(-.78)}$ | $1.27^{(-.11)}$ | $95.01^{(+.71)}$ | $0.23^{(-.03)}$ |
| **Ours** | $\mathbf{5.80}^{(-.63)}$ | $\mathbf{1.12}^{(-.14)}$ | $\mathbf{95.91}^{(+.76)}$ | $\mathbf{0.19}^{(-.03)}$ |
| **Breaking Bad (Artifact) subset** | | | | |
| (Lu et al., 2023) | 43.30 | 7.78 | 64.99 | 8.13 |
| (Wang et al., 2025a) | 46.29 | 9.87 | 60.80 | 7.78 |
| (Xu et al., 2026) | 7.80 | 1.76 | 93.01 | 0.97 |
| (Li et al., 2025) (Mini) | 7.90 | 1.80 | 93.37 | 0.87 |
| (Li et al., 2025) | 5.80 | 1.33 | 94.89 | 0.43 |
| **Ours (Mini)** | $7.11^{(-.79)}$ | $1.37^{(-.43)}$ | $94.22^{(+.85)}$ | $0.68^{(-.19)}$ |
| **Ours** | $\mathbf{5.36}^{(-.44)}$ | $\mathbf{1.01}^{(-.32)}$ | $\mathbf{95.72}^{(+.83)}$ | $\mathbf{0.33}^{(-.10)}$ |
| **FRACTURA (synthetic fractures)** | | | | |
| (Lu et al., 2023) | 62.19 | 18.98 | 32.65 | 71.72 |
| (Wang et al., 2025a) | 60.33 | 17.76 | 39.65 | 46.83 |
| (Li et al., 2025) (Mini) | 27.90 | 6.82 | 76.50 | 7.65 |
| (Li et al., 2025) | 17.98 | 4.50 | 84.18 | 6.36 |
| **Ours (Mini)** | $25.92^{(-1.98)}$ | $5.99^{(-.83)}$ | $78.45^{(+1.95)}$ | $7.02^{(-.63)}$ |
| **Ours** | $\mathbf{17.20}^{(-.78)}$ | $\mathbf{4.03}^{(-.47)}$ | $\mathbf{85.13}^{(+.95)}$ | $\mathbf{5.29}^{(-1.07)}$ |

## 6. Conclusion

Our method is a step towards mathematically grounded generative modeling on $\text{SE}(n)$ shape spaces. Empirically, we observe consistent improvements over baseline methods that do not explicitly handle global rigid-motion ambiguity, most notably in 2D settings where this structure is often overlooked, while still yielding gains in 3D despite a more competitive landscape. Importantly, this framework departs from reliance on restrictive data hypotheses: it is broadly applicable, while providing the mathematical scaffold to integrate domain-specific constraints in future work.

## Impact Statement

This work aims to advance machine learning methods for geometric object reassembly by explicitly accounting for global rigid-motion symmetries. The proposed approach may be useful in applications such as cultural-heritage restoration, geometric part assembly, and fracture reconstruction, where more robust alignment methods can support expert workflows. As with any automated reconstruction method, outputs should be interpreted with care when fragments are ambiguous, noisy, or incomplete, and should be validated by domain experts in real-world use. We do not anticipate direct negative societal impacts beyond those of related geometric reconstruction technologies.

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

# A. Extended Ablation

## A.1. Detailed ablation

The ablation in Tab. 2 shows that enforcing geometric constraints is critical for performance:

- **Global gauge fixing** provides the most significant improvement, increasing Overlap from $0.80$ to $0.90$. By projecting configurations onto a unique canonical section $\Sigma$, the model identifies a single solution within each group orbit, reducing problem complexity. Mathematically, the dimensionality decreases from $\dim(\mathcal{M}) = N \dim(\mathrm{SE}(n))$ to $\dim(\Sigma) = (N - 1) \dim(\mathrm{SE}(n))$, effectively removing the three (in 2D) or six (in 3D) global degrees of freedom.

- **Quotient-invariant loss** allows us to achieve the highest Overlap ($0.93$): by supervising strictly on the horizontal tangent bundle, the network learns intrinsic shape evolution while ignoring spurious gradients from unidentifiable global drift.

- Applying **inference projection** $\Pi_{\mathcal{H}}$ during testing ensures physical consistency. By enforcing zero global momentum throughout the integration process, Overlap reaches $0.94$.

- Even if **multi-anchor** supervision (Row D) improves the unconstrained baseline ($0.87$ vs $0.80$ Overlap), the gap with our Global Gauge Fixing (Overlap $0.90$) highlights a limitation of anchoring. Moreover, GGF allows for faster training dynamics: models utilizing the gauge converge within $1{,}000$ epochs, whereas the multi-anchor baseline requires approximately $1{,}500$ epochs to plateau. This suggests that GGF leads to a sharper, less noisy regression target, effectively simplifying the optimization landscape compared to the combinatorial nature of multi-anchor supervision.

## A.2. Alternative Gauge Selection

We discuss the choice of canonical section $\Sigma$ by comparing a **PCA gauge** to a **Riemannian Fréchet mean gauge**. Both define valid quotient maps $\Psi : \mathcal{M} \to \Sigma$ that fix the $\mathrm{SE}(n)$ degrees of freedom, but they may differ in numerical stability during training. Here, we focus on **Fréchet-mean gauge**, that defines the canonical orientation $R^*$ as the Fréchet mean of the fragment rotations, found by minimizing $\sum d^2_{SO(n)}(R^*, R_i)$, via an iterative algorithm.

As shown in Table 8, PCA is (very) slightly but consistently better on SVJP across B/C, with small yet repeatable margins.

*Table 8.* SVJP gauge-selection ablation. For each setup (B/C), we swap only the gauge. Parentheses report signed differences w.r.t. PCA under the same setup; best values are in **bold**.

| Train config. | Components | | Metrics (vs. PCA) | | |
|---|---|---|---|---|---|
| | Gauge | Inv. loss | O↑ | P↑ | R↑ |
| **(B) + Gauge** | | | | | |
| | PCA | ✗ | $.90^{(+.00)}$ | $.89^{(+.00)}$ | $.88^{(+.00)}$ |
| | Fréchet-mean | ✗ | $.89^{(-.01)}$ | $.89^{(-.00)}$ | $.88^{(-.00)}$ |
| **(C) + Quot. loss** | | | | | |
| | PCA | ✓ | $\mathbf{.93}^{(+.00)}$ | $\mathbf{.90}^{(+.00)}$ | $\mathbf{.89}^{(+.00)}$ |
| | Fréchet-mean | ✓ | $.92^{(-.01)}$ | $.89^{(-.01)}$ | $.88^{(-.01)}$ |

**PCA gauge tie-breaking (deterministic canonicalization).** The PCA gauge is defined by eigenvectors of the positional covariance and is therefore only determined up to sign flips and axis permutations. We resolve these ambiguities deterministically as follows: (i) sort axes by decreasing eigenvalue; (ii) enforce a right-handed frame by flipping one axis if needed to ensure $\det(R^*) = +1$; and (iii) in degenerate cases (eigenvalue multiplicity) where the in-plane axes are not uniquely defined, we select a consistent completion using fragment orientations, as follows.

In 3D, after obtaining the principal axis $e_1$, we exploit rotations by considering $v_i^{(k)} = (I - e_1 e_1^\top) R_i e_k$ for $k \in \{1, 2, 3\}$ and the weights $w_i = \|p_i^c\|$. We set $b_k = \sum_i w_i v_i^{(k)}$ and choose $k^\star = \arg\max_k \|b_k\|$. Defining $s_i = \mathrm{sign}(\langle v_i^{(k^\star)}, b_{k^\star} \rangle)$, we can compute $b = \sum_i w_i s_i v_i^{(k^\star)}$ and $e_2 = \frac{b}{\|b\|}$ (provided that $\|b\|$ is above a small threshold $\tau_b$) and finally we set $e_3 = e_1 \times e_2$. If $\|b\| < \tau_b$, we complete the frame by choosing $a \in \{(1,0,0), (0,1,0), (0,0,1)\}$ (canonical base) such that $|a^\top e_1|$ is minimal (i.e., $a$ is as orthogonal as possible to $e_1$), then setting $e_2 = \frac{e_1 \times a}{\|e_1 \times a\|}$ and $e_3 = e_1 \times e_2$ (with a final sign flip if needed to enforce $\det(R^*) = +1$).

**Empirical analysis on geometric degeneracies.** While perfectly symmetric spectra are rare in 2D settings or in real 3D data, near-degeneracies can arise in simulated datasets containing objects with rotational symmetries or elongated structures. We report results on the subset of Breaking Bad (Sellán et al., 2022) that specifically requires PCA gauge disambiguation. According to Table 9, performance on this subset is even higher than on the full distribution (Tab. 7; e.g. 96.44 vs. 95.91 for PA-Everyday). This should be interpreted with care: the "degenerate" subset is biased toward simpler instances, since PCA ambiguities are much more likely when objects split into only a few large fragments. Consistently, 80% of the samples in this subset contain five fragments or fewer.

*Table 9.* 3D quantitative results (GARF protocol) on the **ambiguous subsets** of Breaking Bad Everyday/Artifact. We report $\text{RMSE}(R^\circ)$ (deg)↓, $\text{RMSE}(T)$ ↓ ($10^{-2}$), part accuracy (PA)↑, and Chamfer distance (CD)↓ ($10^{-3}$). Best values are in **bold**; second-best are underlined.

| Methods | $\text{RMSE}(R^\circ) \downarrow$ | $\text{RMSE}(T) \downarrow$ | PA ↑ | CD ↓ |
|---|---|---|---|---|
| **Breaking Bad (Everyday) - Ambiguous Subset** | | | | |
| GARF (Li et al., 2025) | 5.90 | 1.18 | 96.27 | 0.19 |
| **Ours** | **5.45** | **1.05** | **96.44** | **0.18** |
| **Breaking Bad (Artifact) - Ambiguous Subset** | | | | |
| GARF (Li et al., 2025) | 5.54 | 1.30 | 95.20 | 0.40 |
| **Ours** | **5.02** | **0.99** | **96.03** | **0.31** |

## A.3. Metric Sensitivity and Statistical Stability

We additionally evaluate the sensitivity of our method to the relative weighting between translation and rotation in the tangent-space metric. We consider the weighted metric

$$\|\dot{x}\|_{M_a}^2 = 2 \sum_i \left( a\|\dot{p}_i\|^2 + (1-a)\|\omega_i\|^2 \right), \qquad 0 \le a \le 1,$$

and report results for several values of the ratio $a/(1-a)$. As shown in Table 10, the default balanced setting performs best or near-best across all tested benchmarks, and performance remains stable for moderate changes of the translation–rotation weighting.

*Table 10.* Sensitivity to the translation–rotation weighting in the tangent-space metric.

| $a/(1-a)$ | SVJP Overlap↑ | RePAIR Overlap↑ | B. Bad Everyday PA↑ | FRACTURA PA↑ |
|---|---|---|---|---|
| 0.50 | .91 | .37 | 95.34 | 84.72 |
| 0.75 | .93 | .39 | 95.43 | 85.01 |
| 1.00 (default) | .93 | .41 | 95.91 | 85.13 |
| 1.25 | .92 | .37 | 95.89 | 84.89 |
| 1.50 | .92 | .35 | 93.62 | 83.82 |

We also evaluate the statistical stability of the 3D results over five independent training seeds, using fixed sampling settings. Table 11 reports mean and standard deviation (across five training seeds) on the three 3D benchmarks. The standard deviations are small compared to the observed gains over the strongest baselines, indicating that the reported improvements are stable.

*Table 11.* Statistical stability of our 3D results over five training seeds. We report mean $\pm$ standard deviation across five training seeds.

| Dataset | $\text{RMSE}(R^\circ) \downarrow$ | $\text{RMSE}(T) \downarrow$ | PA↑ | CD↓ |
|---|---|---|---|---|
| Breaking Bad (Everyday) | $5.78 \pm 0.02$ | $1.10 \pm 0.04$ | $95.96 \pm 0.07$ | $0.20 \pm 0.01$ |
| Breaking Bad (Artifact) | $5.35 \pm 0.03$ | $1.00 \pm 0.01$ | $95.74 \pm 0.02$ | $0.31 \pm 0.02$ |
| FRACTURA | $17.10 \pm 0.12$ | $3.98 \pm 0.03$ | $85.15 \pm 0.04$ | $5.12 \pm 0.10$ |

# B. Architecture

### B.1. Shape encoder

PuzzleFusion reports that vertex-level attention can be more accurate in their setting (Hossieni et al., 2023). However, global attention over raw contour vertices scales quadratically with the total vertex count, which quickly becomes prohibitive as contour resolution increases. We therefore adopt a piece-level design: each fragment contour is encoded into a compact token in a d-dimensional latent space and attention is applied across fragments. Despite this compression, we obtain strong performance in our benchmarks.

### B.2. SVJP encoder.

For each polygon $P_i = \{p_{i,m}\}_{m=1}^{M_i}$, the ordered vertex sequence (defining the contour of a piece) is centered ($\forall m \in [\![1, M_i]\!], p_{i,m}^c = p_{i,m} - \frac{1}{M_i} \sum_{j=1}^{M_i} p_{i,j}$) and embedded into a fixed-dimensional piece token of dimension $M$ using zero padding if necessary. Concretely, the ordered vertex sequence is stacked into a fixed-length representation and projected by a fully-connected (linear) layer $\Phi : \mathbb{R}^{2M} \to \mathbb{R}^d$ to obtain:

$$h_i = \Phi\left(\text{vec}(P_i^c)\right) \in \mathbb{R}^d, \tag{16}$$

where $\text{vec}(\dot{)}$ is the flattening operator. This provides a translation-invariant description of the local geometry by operating in the centered local frame, yielding one token per fragment.

### B.3. Polygonal Room Encoder (MagicPlan, RPLAN)

Each room $R_i$ is represented as a sequence of $M$ vertices. To encode the geometry and architectural features jointly, we define a unified feature tuple for the $m$-th vertex as $u_{i,m} = [x_{i,m}, y_{i,m}, \delta_{i,m}] \in \mathbb{R}^3$, where $\delta_{i,m} \in \{0, 1\}$ acts as a binary indicator, taking the value 1 if the vertex belongs to a doorway and 0 otherwise.

The room's geometric representation $v_i$ is derived by flattening previous tuples into a single vector of fixed dimension $3M$:

$$v_i = [x_{i,1}, y_{i,1}, \delta_{i,1}, \ldots, x_{i,M}, y_{i,M}, \delta_{i,M}]^\top \in \mathbb{R}^{3M}. \tag{17}$$

Finally, this geometric vector is concatenated with the semantic one-hot vector $s_i \in \{0,1\}^{20}$ (representing the room category) to form the complete room embedding input:

$$z_i = \begin{bmatrix} v_i, s_i \end{bmatrix} \in \mathbb{R}^{3M+20}. \tag{18}$$

The final room embedding $h_i$ is obtained through a fully connected layer $\Phi : \mathbb{R}^{3M+20} \to \mathbb{R}^d$:

$$h_i = \Phi(z_i). \tag{19}$$

### B.4. SVFP shape encoder.

First, to eliminate the artificial cyclic ambiguity of polygon representations, we pre-process each piece into a canonical vertex order using a deterministic phase criterion. For each polygon $P_i = \{p_{i,m}\}_{m=1}^{M}$ (centered in its local frame), we compute the polar phase:

$$\phi_m = \text{atan2}(p_{i,m}^{(y)}, p_{i,m}^{(x)}) \in [0, 2\pi). \tag{20}$$

We then choose the vertex $m^\star = \arg\min_m \phi_m$ and circularly shift the indexing so that the sequence starts at $p_{m^\star}$. Finally, we fix the traversal direction by applying a deterministic reversal of the shifted order, yielding an equivariant representation up to rigid motion and removing both the cyclic shift and orientation ambiguities. The same re-indexing is applied to any per-vertex attributes.

We then extract translation-invariant local features $x_{i,j} \in \mathbb{R}^4$ for every vertex $j$ of fragment $i$. These features encode the local curvature via the turning angle $\alpha_{i,j}$ and edge lengths $e_{i,j}$:

$$\begin{aligned}
\alpha_{i,j} &= \text{atan2}(\det[e_{i,j-1}, e_{i,j}], \langle e_{i,j-1}, e_{i,j} \rangle), \\
x_{i,j} &= [\cos \alpha_{i,j}, \sin \alpha_{i,j}, |e_{i,j}|, |\bar{v}_{i,j} - \bar{v}_i|].
\end{aligned} \tag{21}$$

To respect the closed-chain topology, we inject a cyclic positional encoding $PE_{cyc}$ composed of Fourier features with frequencies $l = 1, \ldots, L$:

$$PE_{cyc}(j) = \left[ \sin\left(\frac{2\pi lj}{M}\right), \cos\left(\frac{2\pi lj}{M}\right) \right]_{l=1}^{L}. \tag{22}$$

Rather than applying global vertex attention, we process the entire contour with a local Transformer and pool its outputs into a single fragment token:

$$h_i = \text{Pool}\left( T_{\text{local}}\left( \{x_{i,j} + PE_{cyc}(j)\}_{j=1}^{M} \right) \right). \tag{23}$$

### B.5. Piece interaction coding.

Global reasoning operates on piece tokens $\{h_i\}_{i=1}^{N}$ after they have been "augmented" with pose and the time embedding $e(t)$ of the generative process as follows. Let the fragment pose be parametrized as:

$$p_i = [x_i, y_i, \cos\theta_i, \sin\theta_i] \in \mathbb{R}^4 \tag{24}$$

Then, it is projected by a fully connected layer $\text{FC} : \mathbb{R}^4 \to \mathbb{R}^d$. A set-level attention module is then applied over $\{z_i\}_{i=1}^{N}$ where:

$$z_i = h_i + \text{FC}(p_i) + e(t) \tag{25}$$

and the resulting updates are iteratively refined to produce the final $SE(2)$ motion field for all pieces.

## C. Dataset Descriptions and Experimental Context

This section provides the specific details for each experimental setup used in the main paper. Our framework is evaluated across diverse reassembly tasks, ranging from 2D puzzles to 3D artifact reconstruction.

### C.1. 2D Reassembly Benchmarks

#### C.1.1. 2D SYNTHETIC PUZZLES: SVJP

The **Square Voronoi Jigsaw Puzzle (SVJP)** dataset evaluates geometric reasoning under complex, low informative boundaries and variable fragment counts. For each puzzle, we sample $N$ points inside a [0,1] square and construct the corresponding Voronoi tessellation. The resulting Voronoi cells serve as puzzle pieces. The number of fragments ranges from 3 to 15 uniformly over 100,000 puzzles for training and 1,000 puzzles for testing. To assess robustness to imperfect polygon extraction, we additionally evaluate on corrupted inputs obtained by perturbing corner coordinates with i.i.d. Gaussian noise at three levels, using variances of $0.5\%$ and $1\%$ of the square's width. Models are trained on the noise-free setting and evaluated without finetuning under increased noise. We report *Overlap* (average IoU of predicted pieces with ground truth) and *Precision/Recall* computed on the adjacency graph induced by neighboring piece connectivity.

#### C.1.2. 2D SYNTHETIC PUZZLES: SVFP

The **Square Voronoi Fractal Puzzle (SVFP)** dataset follows the same Voronoi-based construction as SVJP. To emulate realistic break contours while preserving exact mutual compatibility, SVFP replaces straight shared edges with *shared fractal curves*: for every unique boundary segment between two adjacent cells, we recursively subdivide the segment for a fixed number of levels and perturb midpoints along the local normal with an amplitude proportional to edge length (with a minimum amplitude and geometric decay), and we reuse the exact same perturbed polyline for both incident pieces. In addition, we optionally generate an *eroded* variant by carving a thin band around the shared curves and subtracting it from each piece, creating small residual gaps that mimic worn borders and imperfect fits. For learning and evaluation, piece polygons are resampled to a fixed number of contour points per piece (e.g., $M = 100$).

#### C.1.3. FLOOR PLAN ASSEMBLY: RPLAN

The RPLAN dataset (Wu et al., 2019) consists of residential floorplans and evaluates constrained assembly where geometry must be consistent with semantically plausible room adjacencies. RPLAN contains 60,000 samples, split into 55,000 training and 5,000 testing floorplans. In the task, each sample provides a shuffled set of room polygons (3 to 8 rooms per floor). Each room is associated with a room-type label encoded as a 20D one-hot vector. A door is represented as a line segment with two corners and is treated as an additional element in the input.

### C.1.4. REAL-WORLD FLOOR PLAN: MAGICPLAN

The MagicPlan dataset (Hossieni et al., 2023) provides a large-scale real-world benchmark derived from consumer reconstructions, introducing realistic noise and annotation artifacts. It contains 98,780 single-story houses/apartments, split into 93,780 training and 5,000 testing samples. Room shapes are reconstructed by users (via corner clicking) through an augmented-reality application, then Manhattan-rectified, and finally manually arranged into a floorplan that serves as ground truth; this pipeline introduces realistic geometric noise compared to synthetic settings. The number of rooms per house ranges from 3 to 10.

### C.1.5. ARCHAEOLOGICAL FRESCO: REPAIR

We evaluate archaeological reassembly on the RePAIR benchmark (Tsesmelis et al., 2024), which contains realistic 2D (and 3D) fresco fragments exhibiting irregular break geometry, erosion, and missing pieces, together with high-resolution imagery and archaeologist-annotated ground-truth poses/metadata. Following the experimental protocol of (Islam et al., 2025), we use the official 2D split comprising 121 puzzles (97 train / 24 test), with 957 fragments in total and report results on the held-out test set.

**Semi-synthetic pretraining.**    When pretraining on semi-synthetic fresco puzzles, we follow the data-construction protocol of (Islam et al., 2025) and build upon the fragmentation procedure of (Zhou et al., 2024), where frescoes are partitioned into fragments by iteratively sampling segmentation lines whose boundaries randomly mix straight segments and Fourier-synthesized curves to yield diverse, realistic break patterns. As in (Islam et al., 2025), we further incorporate border degradation (morphological erosion) and small rigid perturbations to better match RePAIR acquisition artifacts.

## C.2. 3D Reassembly Benchmarks and Protocol

### C.2.1. SYNTHETIC FRACTURES: BREAKING BAD

Breaking Bad (Sellán et al., 2022) is a large-scale synthetic dataset for 3D fracture and reassembly. We use the volume-constrained benchmark protocol and report results separately on two subsets: an Everyday subset with 7,872 assemblies used for evaluation, and an Artifact subset with 3,697 assemblies used for evaluation, designed to test generalization to different object shapes. The benchmark spans a wide range of fragment counts per object, with fractures generated by simulation, providing a controlled environment for quantitative evaluation at scale.

### C.2.2. MIXED SYNTHETIC/REAL SCIENTIFIC FRACTURES: FRACTURA

FRACTURA (Li et al., 2025) is curated to capture real-world fracture complexity across multiple scientific domains and includes both real fracture test data and synthetic fracture counterparts generated from intact objects for controlled experimentation and domain adaptation. The dataset spans several object categories, including ceramics, bones (e.g., vertebrae, limbs, ribs), eggshells, and lithics. However, only the bones dataset is publicly available. The real-fracture subset covers three characteristic fracture regimes: (i) *random breakage*, corresponding to irregular and chaotic fractures commonly observed in ceramics, bones, and eggshells; (ii) *incomplete ossification*, where unfused juvenile bone ends yield fragmented epiphyses; and (iii) *flintnapping*, producing conchoidal fractures in lithics with radially propagating fracture lines. For the synthetic counterpart, fracture generation relies on physics-based simulation for ceramics, bones, and eggshells, and on geometry-based simulation for lithics. Real fragments and intact objects are digitized via high-accuracy 3D scanning: the real-fracture subset is used exclusively for testing, while scans of intact objects support the synthetic fracture generation pipeline. Overall, FRACTURA contains 9,727 synthetic assemblies (53,350 pieces) and 41 real assemblies (292 pieces), with per-category statistics reported in the accompanying appendix of the reference protocol.

## C.3. Hyperparameters

For a fair comparison, we ensure that neural network complexity is comparable across competitors. For experiments on SVJP, SVFP, MagicPlan, and RPLAN, we set the hidden dimension to $d = 256$ with 6 layers of multi-head attention, resulting in an architecture of approximately 5 million parameters similar to PuzzleFusion (Hossieni et al., 2023). When leveraging backbones from other methods, we adopt the exact parameters and official implementations as specified in the original articles.

*Table 12.* Hyperparameters for all experiments (Section C.3), consolidating training configurations, inference dynamics, and hardware details.

| Parameter | 2D Experiments (SVJP, SVFP, RPLAN, MagicPlan, Re-PAIR) | 3D Experiments (Breaking Bad, FRACTURA) |
|---|---|---|
| *Optimization* | | |
| Optimizer | AdamW | AdamW |
| Learning Rate | $10^{-4}$ | $5 \times 10^{-4}$ |
| Batch Size | 128 | 128 |
| Epoch Number | 3000 | 1500 |
| Training time | 0.5 days | 3 days |
| *Inference (Sampling)* | | |
| ODE Integrator | Heun's Method | Heun's Method |
| Integration Steps | 50 | 50 |
| *Infrastructure* | | |
| Hardware | 1 NVIDIA H100 80GB | 4 NVIDIA H100 80GB |

# D. Algorithms

Algorithm 1 describes the training procedure for quotient-invariant flow matching. Given gauge-fixed endpoints $\tilde{x}_0 = \Psi(x_0)$ and $\tilde{x}_1 = \Psi(x_1)$, it samples $t \sim \mathcal{U}[0, 1]$, evaluates the geodesic state $x_t$ and target velocity $u_{\text{tgt}}$, and optimizes $v_\theta$ with the horizontal loss. The required horizontal projection $\Pi_{H_{x_t}}$ is computed by the inertia–momentum solve, with closed forms in the canonical gauge for SE(2) and SE(3).

---

**Algorithm 1** Quotient-Invariant Flow Matching Training on $\mathcal{M} = \text{SE}(n)^N$

---

1: **Input:** data distribution $q$ on $\mathcal{M}$, prior $p_0$ on $\mathcal{M}$, gauge map $\Psi : \mathcal{M} \to \Sigma$, metric matrix $\mathbf{M}$, network $v_\theta$.
2: Sample $x_1 \sim q$,    $\tilde{x}_1 \leftarrow \Psi(x_1)$                                    {Gauge-fix data}
3: Sample $x_0 \sim p_0$,    $\tilde{x}_0 \leftarrow \Psi(x_0)$                                    {Gauge-fix noise}
4: Sample $t \sim \mathcal{U}[0, 1]$
5: $x_t \leftarrow \Gamma_t(\tilde{x}_0, \tilde{x}_1)$                                    {Geodesic interpolant on $\mathcal{M}$ (Eq. 6)}
6: $u_t \leftarrow u_{\text{tgt}}(\tilde{x}_0, \tilde{x}_1)$                                    {Conditional target velocity (Eq. 7)}
7: $\delta \leftarrow v_\theta(x_t, t) - u_t$                                    {Raw residual in $T_{x_t}\mathcal{M}$}
8: $\eta^\star \leftarrow \left( \alpha_{x_t}^\top \mathbf{M} \alpha_{x_t} \right)^{-1} \alpha_{x_t}^\top \mathbf{M} \delta = I(x_t)^{-1} P(x_t)$                                    {Inertia–momentum solve (Eq. 9)}
9: $r \leftarrow \Pi_{H_{x_t}}(\delta) = \delta - \alpha_{x_t}(\eta^\star)$                                    {Horizontal residual (Eq. 8)}
10: $\mathcal{L}(\theta) \leftarrow \|r\|_{\mathbf{M}}^2$                                    {Quotient-invariant FM loss (Eq. 10)}
11: Update $\theta$ using $\nabla_\theta \mathcal{L}(\theta)$

---

Algorithm 2 reports inference on $\text{SE}(n)^N$: it integrates the projected ODE $\dot{x} = \Pi_{H_x}(v_\theta(x, t))$ using Heun steps on the manifold via Exp, optionally re-applying $\Psi$ to limit gauge drift.

---

**Algorithm 2** Sampling with Horizontal Projection on $\mathcal{M} = \mathrm{SE}(n)^N$

---

1: **Input:** trained field $v_\theta$, prior $p_0$, gauge map $\Psi$, time grid $\{t_k\}_{k=0}^K$ with $t_0 = 0, t_K = 1$, step size $\Delta t$.
2: **Output:** $\hat{x} \in \Sigma$ (gauge-fixed assembled configuration)
3: Sample $x_0 \sim p_0, \quad x_0 \leftarrow \Psi(x_0)$                                                {Initialize on the canonical section $\Sigma$}
4: **for** $k = 0$ **to** $K - 1$ **do**
5:        $w_k \leftarrow v_\theta(x_k, t_k)$
6:        $w_k \leftarrow \Pi_{H_{x_k}}(w_k)$                                             {Remove instantaneous global drift (Eq. 15)}
7:        $\tilde{x} \leftarrow \mathrm{Exp}_{x_k}(\Delta t\, w_k)$
8:        $\tilde{w} \leftarrow v_\theta(\tilde{x}, t_{k+1})$
9:        $\tilde{w} \leftarrow \Pi_{H_{\tilde{x}}}(\tilde{w})$
10:      $x_{k+1} \leftarrow \mathrm{Exp}_{x_k}\left(\frac{\Delta t}{2}(w_k + \tilde{w})\right)$
11:      $x_{k+1} \leftarrow \Psi(x_{k+1})$                                                {Gauge re-fix for numerical stability}
12: **end for**
13: **return** $x_K$

---

## E. Detailed Mathematical Derivations

In this section, we provide the rigorous derivation of the horizontal projection operator $\Pi_{\mathcal{H}_x}$ presented in Section 4. We show that enforcing zero global drift corresponds to solving a generalized inertia-momentum problem on the Lie algebra $\mathfrak{se}(n)$.

### E.1. The Horizontal Projection Problem

Let $x = (g_1, \ldots, g_N) \in \mathcal{M}$ be a configuration in the product manifold $\mathcal{M} = \mathrm{SE}(n)^N$. We consider a raw velocity residual $\delta = (\delta_1, \ldots, \delta_N) \in \mathfrak{g}^N$, typically defined as the difference between the network prediction and the target field, $\delta = v_\theta(x, t) - u_t(x)$.

The vertical space $\mathcal{V}_x \subset T_x\mathcal{M}$ consists of velocities induced by a global rigid motion $\eta \in \mathfrak{se}(n)$ applied to the entire assembly. For a global spatial twist $\eta$, the induced velocity on each fragment $i$ is given by the infinitesimal action $\alpha_x(\eta)$. The horizontal projection seeks to find the optimal global correction $\eta^*$ that minimizes the residual energy weighted by the metric. Using the mass matrix $\mathbf{M} = \mathrm{diag}(\mathbf{I}_n, \lambda \mathbf{I}_{\dim \mathfrak{so}(n)})$, we define the objective:

$$\min_{\eta \in \mathfrak{se}(n)} \mathcal{L}(\eta), \quad \text{where} \quad \mathcal{L}(\eta) = \frac{1}{2}\sum_{i=1}^{N} \|\delta_i - \alpha_{x,i}(\eta)\|_{\mathbf{M}}^2 \tag{26}$$

### E.2. Solving the Normal Equations

Using the property of the metric-weighted inner product $\langle u, v \rangle_{\mathbf{M}} = u^T \mathbf{M} v$, we expand the loss function:

$$\mathcal{L}(\eta) = \frac{1}{2}\sum_{i=1}^{N} (\delta_i - \alpha_{x,i}(\eta))^T \mathbf{M}(\delta_i - \alpha_{x,i}(\eta)) \tag{27}$$

$$= \frac{1}{2}\sum_{i=1}^{N} \left[\delta_i^T \mathbf{M}\delta_i - 2\delta_i^T \mathbf{M}\alpha_{x,i}(\eta) + \alpha_{x,i}(\eta)^T \mathbf{M}\alpha_{x,i}(\eta)\right] \tag{28}$$

Setting the gradient with respect to the global twist $\eta$ to zero yields:

$$\nabla_\eta \mathcal{L}(\eta) = \sum_{i=1}^{N} \left[-\alpha_{x,i}^T \mathbf{M}\delta_i + \left(\alpha_{x,i}^T \mathbf{M}\alpha_{x,i}\right)\eta\right] = 0 \tag{29}$$

This yields the linear system $\mathcal{I}(x)\eta^* = \mathcal{P}(x)$, which is the generalized form of the **Inertia-Momentum equations**:

$$\mathcal{I}(x) = \sum_{i=1}^{N} \alpha_{x,i}^T \mathbf{M} \alpha_{x,i} \in \mathbb{R}^{\dim \mathfrak{g} \times \dim \mathfrak{g}} \quad \text{(Generalized Inertia Tensor)} \tag{30}$$

$$\mathcal{P}(x) = \sum_{i=1}^{N} \alpha_{x,i}^T \mathbf{M} \delta_i \in \mathbb{R}^{\dim \mathfrak{g}} \quad \text{(Generalized Total Momentum)} \tag{31}$$

Under non-degenerate configurations, $\mathcal{I}(x)$ is symmetric positive definite and thus invertible. The horizontal projection of the residual is then:

$$\Pi_{\mathcal{H}_x}(\delta) = \delta - \alpha_x(\eta^*), \quad \text{where} \quad \eta^* = \mathcal{I}(x)^{-1}\mathcal{P}(x) \tag{32}$$

## F. Notation Summary

*Table 13.* Notation used in Sec. 4.

| Notation | Meaning |
|---|---|
| **Basic groups and dimensions** | |
| $n$ | Ambient dimension ($n = 2$ or $n = 3$ in experiments). |
| $N$ | Number of rigid parts (fragments) in the assembly. |
| $\mathrm{SE}(n)$ | Special Euclidean group in dimension $n$ (rigid motions: translation + rotation). |
| $SO(n)$ | Special orthogonal group (rotations in $\mathbb{R}^n$). |
| $\mathfrak{so}(n)$ | Lie algebra of $SO(n)$ (skew-symmetric matrices; angular velocities). |
| $\mathfrak{se}(n)$ | Lie algebra of $\mathrm{SE}(n)$ (infinitesimal rigid motions). |
| **Configuration space and quotient** | |
| $G$ | Shorthand for the rigid-motion group $G = \mathrm{SE}(n)$. |
| $\mathcal{M}$ | Multi-body configuration manifold: $\mathcal{M} := G^N = \mathrm{SE}(n)^N$. |
| $x$ | Assembly configuration $x = (g_1, \dots, g_N) \in \mathcal{M}$. |
| $g_i$ | Pose of fragment $i$: $g_i = (\mathbf{p}_i, R_i) \in \mathrm{SE}(n)$. |
| $\mathbf{p}_i$ | Position (translation) of fragment $i$ in $\mathbb{R}^n$. |
| $R_i$ | Orientation (rotation) of fragment $i$ in $SO(n)$. |
| $h \cdot x$ | Diagonal *left* action of $h \in \mathrm{SE}(n)$: $h \cdot x = (hg_1, \dots, hg_N)$. |
| $\mathcal{S}$ | Relative-pose (shape) space as a quotient: $\mathcal{S} := \mathcal{M}/G$. |
| **Tangent spaces, coordinates, and metric** | |
| $T(\cdot), T_x(\cdot)$ | Tangent bundle, and tangent space at $x$ (e.g., $T\mathcal{M}, T_x\mathcal{M}$). |
| $\dot{x}$ | Tangent vector (velocity) at $x$: $\dot{x} = (\dot{g}_i)_{i=1}^N \in T_x\mathcal{M}$. |
| $\dot{g}_i$ | Velocity of fragment $i$: $\dot{g}_i = (\dot{\mathbf{p}}_i, \dot{R}_i) \in T_{g_i}\mathrm{SE}(n)$. |
| $\boldsymbol{\omega}_i$ | Angular-velocity coordinates for fragment $i$ (in $\mathbb{R}^{\dim SO(n)}$). |
| $\widehat{(\cdot)}$ | "Hat" map from angular-velocity coordinates to $\mathfrak{so}(n)$, defined by $\dot{R}_i = \widehat{\boldsymbol{\omega}}_i R_i$. |
| $\mathbf{M}$ | Block-diagonal metric matrix used in the quadratic norm, typically $\mathbf{M} = \mathrm{diag}(I_n, I_{\dim SO(n)})$. |
| $\|\dot{x}\|_{\mathbf{M}}^2$ | Product metric on $T_x\mathcal{M}$: $\sum_{i=1}^N (\|\dot{\mathbf{p}}_i\|^2 + \|\boldsymbol{\omega}_i\|^2)$. |
| **Vertical/horizontal decomposition (global drift vs. shape change)** | |
| $\mathcal{V}_x$ | Vertical space at $x$: tangent to the $G$-orbit (global rigid-body drift). |
| $\mathcal{H}_x$ | Horizontal space at $x$: $\mathcal{H}_x := \mathcal{V}_x^\perp$ under $\langle \cdot, \cdot \rangle_{\mathbf{M}}$. |
| $\eta$ | Infinitesimal rigid motion $\eta = (\mathbf{v}, \boldsymbol{\omega}) \in \mathfrak{se}(n)$. |
| $\alpha_x$ | Infinitesimal action $\alpha_x : \mathfrak{se}(n) \to T_x\mathcal{M}$ applied identically to all parts: $(\alpha_x(\eta))_i = [\mathbf{v} + \widehat{\boldsymbol{\omega}}\mathbf{p}_i; \ \boldsymbol{\omega}]$. |
| $\Pi_{\mathcal{H}_x}$ | $\mathbf{M}$-orthogonal projection onto $\mathcal{H}_x$ (removes best-fit global drift). |
| **Canonical gauge fixing** | |
| $\Sigma$ | Canonical section (a set intersecting almost every $G$-orbit once). |
| $\Psi$ | Gauge map $\Psi : \mathcal{M} \to \Sigma$ (centering + PCA axis alignment). |
| $\bar{\mathbf{p}}$ | Centroid: $\bar{\mathbf{p}} = \frac{1}{N}\sum_i \mathbf{p}_i$. |
| $\mathbf{p}_i^c$ | Centered position: $\mathbf{p}_i^c = \mathbf{p}_i - \bar{\mathbf{p}}$. |
| $C$ | Empirical covariance of centered positions: $C = \frac{1}{N}\sum_i \mathbf{p}_i^c(\mathbf{p}_i^c)^\top$. |
| $C = U\Lambda U^\top$ | Eigendecomposition of $C$ (used to define principal axes). |
| $R^*$ | Canonical global rotation, typically $R^* = U^\top$ with deterministic sign convention and $\det(R^*) = 1$. |
| $\bar{x} = \Psi(x)$ | Canonicalized configuration (unique representative on $\Sigma$). |
| $\mathbf{p}_i^*, R_i^*$ | Canonicalized pose components: $\bar{\mathbf{p}}_i = R^* \mathbf{p}_i^c, \tilde{R}_i = R^* R_i$. |
| **Geodesic probability path and targets** | |
| $x_0, x_1$ | Noise sample $x_0 \sim p_0$ and data sample $x_1 \sim q$. |
| $\tilde{x}_0, \tilde{x}_1$ | Canonicalized endpoints: $\tilde{x}_0 = \Psi(x_0), \tilde{x}_1 = \Psi(x_1)$. |
| $p_t(x)$ | Probability path defined via geodesic interpolation on $\mathcal{M}$. |
| $x_t$ | Interpolated configuration at time $t \in [0, 1]$ between $\bar{x}_0$ and $\bar{x}_1$. |
| $\exp, \log$ | Lie group exponential / logarithm maps on $SO(n)$ (used for rotation interpolation). |
| $u_t(x_t)$ | Target velocity field (tangent of the interpolant); constant in chosen coordinates along the path. |
| $u_i^{tgt}$ | Per-fragment target velocity: $[\bar{\mathbf{p}}_{1,i} - \bar{\mathbf{p}}_{0,i}; \ \log(\bar{R}_{1,i}\bar{R}_{0,i}^\top)]$. |
| **Flow Matching model, residuals, and quotient-invariant loss** | |
| $v_\theta(x_t, t)$ | Learned velocity field (neural network) at state $x_t$ and time $t$. |
| $\delta$ | Raw residual in $T_{x_t}\mathcal{M}$: $\delta = v_\theta(x_t, t) - u_t(x_t)$. |
| $\delta_{\mathbf{v},i}, \delta_{\boldsymbol{\omega},i}$ | Translational and rotational components of the residual for fragment $i$. |
| $\eta^*$ | Best-fit global drift (least squares): $\eta^* = \arg\min_{\eta \in \mathfrak{se}(n)} \|\delta - \alpha_{x_t}(\eta)\|_{\mathbf{M}}^2$. |
| $\mathcal{I}(x)$ | Normal-matrix term (information / inertia-like): $\mathcal{I}(x) = \alpha_x^\top \mathbf{M} \alpha_x$. |
| $\mathcal{P}(x)$ | Right-hand term: $\mathcal{P}(x) = \alpha_x^\top \mathbf{M} \delta$. |
| $\mathcal{L}(\theta)$ | Quotient-invariant FM loss: $\mathbb{E}_{t, x_0, x_1}[\|\Pi_{\mathcal{H}_{x_t}}(\delta)\|_{\mathbf{M}}^2]$. |
| **Closed-form projections (canonical gauge)** | |
| $\bar{\delta}_{\mathbf{v}}$ | Mean translational residual: $\bar{\delta}_{\mathbf{v}} = \frac{1}{N}\sum_i \delta_{\mathbf{v},i}$. |
| $\delta_{\mathbf{v},i}^c$ | Centered translational residual: $\delta_{\mathbf{v},i}^c = \delta_{\mathbf{v},i} - \bar{\delta}_{\mathbf{v}}$. |
| $SE(2): J$ | Planar $\pi/2$-rotation matrix $J = \begin{bmatrix} 0 & -1 \\ 1 & 0 \end{bmatrix}$. |
| $SE(2): I_{\mathrm{rot}}, \tau$ | Scalar inertia-like term and torque-like term used to compute $\omega^* = \tau/I_{\mathrm{rot}}$. |
| $SE(3): \mathbf{J}, \mathbf{L}$ | 3D inertia matrix and angular momentum used to compute $\boldsymbol{\omega}^* = \mathbf{J}^{-1}\mathbf{L}$. |
| **Inference** | |
| $\frac{dx}{dt} = \Pi_{\mathcal{H}_x}(v_\theta(x, t))$ | Sampling ODE: integrate the *projected* learned field to evolve only in shape space. |

