# OpenReview forum: "SE(n)-Invariant Flow Matching: A General Framework with Application to Object Reassembly"
_ICML.cc/2026/Conference — ICML 2026 regular_

### Official Review · Reviewer_EoKq · 2026-03-01

**Soundness:** 3
**Presentation:** 2
**Significance:** 3
**Originality:** 2
**Overall Recommendation:** 5
**Confidence:** 4

**Summary:**

This paper addresses the problem of reassembling $N$ rigid fragments in $n$-dimensional space using a generative flow-matching framework that is, by construction, invariant to global SE($n$) rigid motions. The core insight is that naively training on the product manifold $M$ = SE$(n)^N$ is ill-posed because many configurations are equivalent up to a global rigid transform, and the resulting "vertical drift" wastes model capacity and destabilizes training. The authors propose two coupled mechanisms: (1) Global Gauge Fixing (GGF) — a deterministic PCA-based canonicalization that maps each configuration to a canonical representative; and (2) a quotient-invariant Flow Matching loss that projects velocity residuals onto the horizontal tangent bundle via a closed-form inertia-momentum solve, ensuring gradients reflect only shape-changing dynamics. The approach is evaluated on 2D jigsaw/fractal puzzles, floor-plan assembly, archaeological fresco reconstruction, and 3D fracture reassembly, showing consistent improvements over anchoring-based baselines.

**Compliance With Llm Reviewing Policy:**

Affirmed.

**Final Justification:**

I have updated my record, based on authors response to my initial review.

**Key Questions For Authors:**

1. How sensitive are results to the relative weighting of translation vs. rotation in the metric $M$? Have you experimented with a data-driven or learned length scale, or with normalization strategies when object scales vary?

2. The PCA gauge can be discontinuous near eigenvalue multiplicities. Did you observe training instabilities in such cases?

3. Can you provide 3D ablations isolating (i) gauge fixing only, (ii) quotient-invariant loss only, (iii) test-time projection?

4. Does the quotient-invariant loss preserve the CFM “correct gradient” property on the quotient space $S$, or can you formalize conditions under which minimizing Eq. (10) is equivalent to $FM$ on $S$?

5. What is the runtime overhead of the horizontal projection per step during training and sampling?

6. How does the method perform in scenarios with missing/extraneous pieces in 3D (as in GARF’s robustness tests)? Does the horizontal projection remain stable when $N$ varies between training and testing?

7. Are the 3D improvements statistically significant across seeds? Please report mean±std over multiple runs or bootstrap confidence intervals.

**Limitations:**

The work does not provide a clear picture of its limitations or impact that explicitly lists where the approach may fail or become unstable.  This limitations may be caused by (i) PCA gauge degeneracies / near-symmetries (eigenvalue multiplicities) and potential discontinuities; (ii) sensitivity to the translation–rotation weighting in the tangent-space metric (what “horizontal” means depends on this choice); (iii) dependence on reasonably accurate fragment geometry—performance may degrade with extreme noise, severe erosion, or highly symmetric assemblies. Please provide an argument as why this are not an issue for your proposal.

The manuscript lacks a discussion of scalability/compute limitations. It needs to report and contextualize training/sampling cost and, in particular, quantify the incremental runtime overhead of the horizontal projection and any re-gauging during sampling.

Finally, it does not clarify applicability limitations. The current 3D evaluation does not cover scenarios with missing/extraneous fragments (common in real fracture/archaeology settings). If the method is intended for such cases, discuss expected behavior and potential failure modes when $N$ varies between train/test.

**Strengths And Weaknesses:**

**Strengths:**

The work introduces a principled quotient-space treatment of global SE(n) symmetry for flow matching: a horizontal-bundle projection that removes vertical (global-drift) components at every time step. It provides efficient, closed-form SE(2)/SE(3) formulas for the projection via an inertia–momentum least-squares fit, enabling simple, differentiable implementations. It also proposes Global Gauge Fixing (canonical section via PCA) to select a single representative per orbit. the work integrates the geometric construction with rectified-flow-style CFM, resulting in straighter, identifiable targets on the quotient. The evaluation focuses across diverse 2D/3D settings (SVJP, SVFP, RPLAN, MagicPlan, RePAIR, Breaking Bad, FRACTURA), with consistent improvements and provides ablations isolating gauge fixing, quotient-invariant loss, and test-time projection in 2D; shows robust gains versus multi-anchor training and augmentation.Finally, it demonstrates compatibility by instantiating the method on top of published backbones, including replacing GARF’s anchor-based training in 3D.

In terms of presentation, figure 1 is an effective visual summary; the orbit/vertical drift problem and the canonical section fix are clear. Table 1 is a genuinely useful taxonomy of symmetry-handling strategies across the reassembly literature. The notation summary (Table 10, App. F) is thorough and substantially aids readability of the dense mathematical sections.


In terms of significance, the work addresses a fundamental symmetry that plagues learning-based reassembly and related pose-generation tasks, replacing ad-hoc anchors/augmentation with a theoretically grounded quotient-space approach. The proposed framework is general and likely to benefit other SE($n$)-symmetric generative tasks beyond reassembly (e.g., multi-body pose generation, robotics).

In terms of originality, the combination of (i) deterministic PCA gauge fixing, (ii) horizontal projection at both training and inference, and (iii) closed-form inertia–momentum solve — deployed together in a unified framework — is a creative synthesis. The quotient-invariant loss is a conceptually clean contribution that separates shape dynamics from global drift by construction rather than approximation.


**Weaknesses:**

PCA-based gauge fixing can be discontinuous near eigenvalue multiplicities. While the supplement provides deterministic tie-breaking and a small empirical check, differentiability and training stability near degeneracies are not fully analyzed. Non-smoothness at degeneracies can, in principle, produce target discontinuities in training; empirically this seems benign, but a short theoretical or empirical stability note would strengthen the case. The quotient-invariant loss replaces standard FM supervision by projecting residuals; theoretical guarantees relating this modified objective to the original FM/CFM optimality (e.g., matching gradients on the quotient) are not formalized. The quotient-invariant loss (Eq. 10) changes the learning objective relative to classical FM/CFM. It is intuitively justified (learn only intrinsic shape evolution) and works well empirically; a brief theoretical note on how this relates to FM on the quotient would clarify guarantees. Most ablations are in 2D; there is no 3D ablation isolating the contributions of gauge fixing vs. horizontal projection when plugged into the GARF backbone. Results report means only (Table 3 averages over 10 runs but does not report std/CI; other tables do not state seeds). Also, runtime/compute overhead from projection during training/inference is not quantified.

There are several issues with the quality of presentation. The several missing: in the manuscript; number of seeds / statistical reporting policy is not stated in the main paper; the "ten runs" for Table 3 appears only in the caption and is not mentioned for Tables 4–6. App. C.3 (hyperparameters) notes "we set the hidden dimension to $d = 256 ...$ resulting in an architecture of approximately 5 million parameters similar to (?)"—a missing citation appears as a question mark. No ablation on the number of ODE integration steps (fixed at 50 Heun steps throughout). Whether the quotient-invariant method converges with fewer steps—a natural benefit one might expect from the reduced regression variance—is not explored. Hyperparameter/metric choices are not justified. The paper uses an implicit equal weighting between translation and rotation in the tangent-space metric (Eq. 2). Since this choice affects what counts as ‘horizontal’ vs ‘vertical,’ a brief sensitivity analysis (e.g., varying relative translation/rotation scaling) would strengthen the claims. The supplementary algorithm descriptions (Algorithms 1–2) are clear and helpful, but Algorithm 2 includes a gauge re-fix step (line 11) that is not analyzed — it is unclear whether this re-fixing introduces bias or how often it is needed. On a more technical side, compute cost is non-trivial (e.g., 3D reported as 3 days on 4×H100) and the runtime overhead of projection/re-gauging vs anchored methods is not quantified.

There are also several issues related to significance of the contribution. The individual components are not individually new: PCA-based alignment is classical; flow matching on SE(3) / manifolds exists (Bose et al., 2024; Chen & Lipman, 2024); horizontal projection via inertia-momentum is known from mechanics (Montgomery, 1993). The novelty lies in the assembly of these components for the reassembly problem. The paper could better articulate what new theoretical insight (beyond the engineering combination) emerges.


**Refernces**
- Bose, Joey, et al. "SE (3)-Stochastic Flow Matching for Protein Backbone Generation." The Twelfth International Conference on Learning Representations (2024).

- Chen, Ricky TQ, and Yaron Lipman. "Flow Matching on General Geometries." The Twelfth International Conference on Learning Representations (2024).

- Montgomery, Richard. "Gauge theory of the falling cat." Fields Inst. Commun 1 (1993): 193-218.

---

> ### Author Rebuttal · Authors · 2026-03-31
>
> We thank you for the constructive review. We will correct the typos and presentation issues (notably weakness 2). Below we address your technical questions. Please let us know if further clarification is needed.
> # Q1:
> We agree this weighting matters.
> Consider the weighted metric $\lVert \dot x \rVert_{M_{a,b}}^2 = 2\sum_i \left(a\lVert \dot p_i \rVert^2 + (1-a)\lVert \omega_i \rVert^2\right), 0 \leq a \leq 1$.
> However, choosing a=.5 does not mean that translation and rotation are treated identically. Their relative influence is further modulated by the quotient projection, which depends on the current fragment configuration through the inertia matrix. In this sense, the method already includes a configuration-dependent normalization rather than a separate learned length scale.
> Empirically, the neutral choice a=.5 performs best across the four tested settings, as reported below:
> |a/(1-a)|SVJP overlap↑|RePAIR overlap↑|B. Bad (Everyday) PA↑|FRACTURA PA↑|
> |-|-|-|-|-|
> |0.50|.91|.37|95.34|84.72|
> |0.75|.93|.39|95.43|85.01|
> |1.00|.93|.41|95.91|85.13|
> |1.25|.92|.37|95.89|84.89|
> |1.50|.92|.35|93.62|83.82|
>
> We therefore did not introduce an additional learned length scale.
> # Q2:
> We agree that PCA-based gauge fixing can be non-smooth near repeated eigenvalues and may in principle create discontinuities in the canonical representative. This is an intrinsic identifiability issue rather than a pathology of our gauge: near-symmetric target configurations (e.g.,colinear barycenters) do not admit a globally smooth deterministic canonicalization. Since $x_0$ is a.s non-degenerate under absolutely continuous sampling, the practically relevant case arises near ambiguous targets $x_1$ at $t \approx 1$ when the interpolant model is already close to the target. Empirically, we observe that our deterministic tie-breaking rule significantly reduces target variability to level observed in non degenerate cases, as it is deterministic. We will add this study in Supplement.
> # Q3:
> We report the requested 3D ablation on Breaking Bad showing that both gauge fixing and the quotient loss contribute:
> |Variant|Everyday||||Artifact||||
> |-|-|-|-|-|-|-|-|-|
> ||RMSE(R)↓|RMSE(T)↓|PA↑|CD↓|RMSE(R)↓|RMSE(T)↓|PA↑|CD↓|
> |No invariant component|9.67|3.23|91.48|0.76|10.02|3.98|90.21|0.98|
> |+ Gauge fixing only|6.98|1.54|94.76|0.26|6.02|1.36|94.93|0.41|
> |+ Quotient loss only|6.07|1.29|95.34|0.22|5.76|1.30|95.01|0.37|
> |+ Gauge fixing + quotient loss|5.74|1.13|95.54|0.19|5.41|1.12|95.23|0.34|
> |+ Gauge fixing + quotient loss + test-time projection|5.80|1.12|95.91|0.19|5.36|1.01|95.72|0.33|
> # Q4:
> Yes. Let $\mu(dt,dx)=dtp_t(dx)$ be the law of $(t,x_t)$. Then Eq. (10) reads $L(\theta)=\int \|\Pi_{H_x}(v_\theta(x,t)-u_t^{\mathrm{tgt}})\|^2_M\,\mu(dt,dx)$. Since for each fixed $x$ the map $\Pi_{H_x}$ is linear, this is equivalently
> $L(\theta)=\int \|\Pi_{H_x}v_\theta(x,t)-\Pi_{H_x}u_t^{\mathrm{tgt}}\|^2_M\,\mu(dt,dx)$.
> Define $A_\theta(x,t):=\Pi_{H_x}v_\theta(x,t)$ and $\bar u_t^H(x):=\mathbb{E}[\Pi_{H_{x_t}}u_t^{\mathrm{tgt}}\mid x_t=x]$. Then conditional bias-variance yields $L(\theta)=\int_0^1 E_{x \sim pt}\big[\|A_\theta(x,t)-\bar u_t^H(x)\|^2_M\big]dt+C$, where $C$ is independent of $\theta$. Hence Eq. (10) preserves the usual CFM $L^2$ regression property after horizontal projection: its minimizer satisfies $A_\theta=\bar u_t^H$ in $L^2$, and Eq.(10) has the same parameter gradient as the corresponding ideal horizontal FM risk. We will add a full theorem and proof in the Supplement.
> # Q5:
> The projection is very efficient: it only adds an $O(N)$ accumulation over fragments and then a small $1\times1$(2D) or $3\times3$(3D) linear solve, so its cost is negligible relative to the backbone. Measured runtimes are:
> |Model|Train step (min/epoch)|Sampling (s) (Batch size=16)|
> |-|-|-|
> |GARF baseline|$2.14\pm0.05$|$3.12\pm0.07$|
> |Ours|$2.15\pm0.04$|$3.17\pm0.06$|
> # Q6:
> From a theoretical standpoint, there is no specific limitation in such cases. We report the following results following GARF settings on Breaking Bad everyday.
> |Protocol|Method|RMSE(R)↓|RMSE(T)↓|PA↑|CD↓|
> |-|-|-|-|-|-|
> |Missing 20% pieces|GARF|23.81|5.12|76.90|1.20|
> |Missing 20% pieces|Ours|20.92|4.18|80.38|1.13|
> |20% extra pieces|GARF|24.91|5.74|79.33|1.01|
> |20% extra pieces|Ours|19.29|3.97|81.43|0.99|
>
> The horizontal projection depends on first/second-order configuration statistics (centroid / inertia), not directly on $N$, so changes in fragment count are not in themselves a source of instability. In our experiments, as usual, training covers varied puzzle sizes and inference is assumed within the same regime.
> # Q7:
> We have evaluated the method over 5 training seeds with fixed sampling settings and report the following results:
> |Dataset|RMSE(R)↓|RMSE(T)↓|PA↑|CD↓|
> |-|-|-|-|-|
> |Breaking Bad (Everyday)|$5.78\pm0.02$|$1.10\pm0.04$|$5.96\pm0.07$|$0.20\pm0.01$|
> |Breaking Bad (Artifact)|$5.35\pm0.03$|$1.00\pm0.01$|$95.74\pm0.02$|$0.31\pm0.02$|
> |FRACTURA|$17.10\pm0.12$|$3.98\pm0.03$|$85.15\pm0.04$|$5.12\pm0.10$|

---

> > ### Author Rebuttal · Reviewer_EoKq · 2026-04-01
> >
> > The authors have addressed all major weaknesses, and the proposed framework represents a principled, efficient, and high-performing solution for $SE(n)$-invariant tasks.

---

### Official Review · Reviewer_MFqn · 2026-03-05

**Soundness:** 3
**Presentation:** 3
**Significance:** 3
**Originality:** 3
**Overall Recommendation:** 5
**Confidence:** 4

**Summary:**

This paper addresses the global rigid-motion ambiguity inherent in multi-part object reassembly, where configurations in $\mathcal{M}=\mathrm{SE}(n)^N$ are identifiable only up to a global $\mathrm{SE}(n)$ transformation. To tackle this, the authors propose: (1) a deterministic Global Gauge Fixing (GGF) canonicalization strategy based on centering and Principal Component Analysis (PCA) alignment; and (2) a quotient-invariant Flow Matching objective that effectively removes global drift at each timestep by projecting residual velocities onto the horizontal space. The proposed approach admits closed-form solutions for both $\mathrm{SE}(2)$ and $\mathrm{SE}(3)$. Comprehensive evaluations on 2D polygonal puzzles, floorplans, the RePAIR archaeological benchmark, and 3D fracture reassembly datasets demonstrate consistent quantitative improvements over several baseline methods.

**Compliance With Llm Reviewing Policy:**

Affirmed.

**Final Justification:**

My problem has been solved and it is a solid work.

**Key Questions For Authors:**

1. I noticed a recent ICLR 2026 paper titled Quotient-Space Diffusion Models. Although that work focuses on diffusion models, it similarly constrains the generative process within the quotient space and leverages the concept of tangent space decomposition. Could the authors discuss the connections and distinctions between their method and this paper? Additionally, I would be interested to know if the model proposed in that work could also be applied to the specific multi-part reassembly scenarios discussed in this submission.
2. Could the authors provide a quantitative comparison of the wall-clock runtimes between the base GARF model and the proposed quotient-invariant version?
3. As discussed in Weakness 2, would alternative metric definitions have a significant impact on the model's overall performance?

**Limitations:**

yes

**Strengths And Weaknesses:**

### **Strengths**
1.  **Solid Theoretical Foundation:** The mathematical framework is concise and rigorous. Leveraging the inertia-momentum equations to analytically project velocity gradients onto the horizontal tangent space provides an elegant, closed-form solution to factor out unidentifiable global pose drift.
2.  **Strong Empirical Performance:** The authors evaluate the framework across an impressively broad range of domains. The consistent performance improvements over competitive, domain-specific baselines provide highly convincing evidence for the utility of the proposed method.
3.  **Architectural Flexibility and Clarity:** The proposed components can be seamlessly integrated into standard flow matching frameworks. Furthermore, the paper is well-presented, organized, and easy to follow.


### **Weaknesses**
1.  **Choice of Metric:** The model relies heavily on the metric defined in Equation 2. However, this definition merely assigns equal weights to the translation and rotation components and sums them up. Since translation and rotation typically possess different physical dimensions, this naive summation may lead to an imbalance between the two during the training process.
2.  **Limited Ablation Coverage:** Table 2 is informative but only for one dataset; it leaves open whether quotient loss and test-time projection matter similarly on 3D or on real RePAIR datasets.
3.  **PCA Instability:** The proposed Global Gauge Fixing (GGF) fundamentally relies on the assumption of distinct eigenvalues. Although the authors provide a tie-breaking workaround for degenerate cases in the appendix, this heuristic approach lacks rigorous theoretical guarantees and could likely lead to training instability. Furthermore, the empirical evaluation of near-degenerate scenarios is overly simplistic. It is insufficient to convincingly demonstrate that the PCA-based method can robustly handle cases where eigenvalues are very close.

---

> ### Author Rebuttal · Authors · 2026-03-31
>
> We thank the reviewer for the careful reading, the clear summary, and the positive recommendation. We are especially glad that the geometric core of the method, projecting out the global rigid drift in tangent space, came across clearly. Below, we address the three questions and briefly clarify the two remaining concerns.
>
> # Q1: Relation to an additional paper:
>
> We thank the reviewer for pointing us to this paper. To our knowledge, it became publicly visible 2 days before the ICML deadline. Hence, this did not leave us enough time to cite it and make a fair comparison during submission. To address the reviewer’s concern, we have now adapted their method to our 3D setting and conducted an empirical comparison under the same backbone, compute budget, and non-anchored protocol. We will add both the citation and this comparison to the revised version if allowed (the paper is officially unpublished and lacks a DOI).
>
> At a high level, the two works are related in spirit but differ substantially in both setting and implementation:
> - Generative paradigm: Their work studies a quotient-space formulation for diffusion; ours is developed in the more recent deterministic flow matching settings. Thus, both methods have a strict non overlapping methodology by design.
> - State space: This paper is not about rigid motion learning. Their framework is designed for Euclidean coordinate generation (e.g,  $R^{3N}/SE(3)$ for 3D point cloud structure like molecular structures), whereas our state space is the product Lie group $SE(n)^N/SE(n)$, which admits explicit closed-form constructions for rigid motions for n =2 and n=3. In particular, our representation preserves all properties of rigid motion, such as the 2$\pi$-periodicity of rotational coordinates, which are essential in rigid reassembly.
> QSDM is therefore not directly plug-and-play for rigid motion prediction, but we agree that an empirical comparison is important.
> To address this, we adapted their training objective to our 3D setting using the same backbone, compute budget, and non-anchored strategy as in our method. Our experiment therefore consists of our 3D setting, with the loss function derived from the one proposed in QSDM in the proposed diffusion setting. For reproducibility, we used the same sampling parameter as the one reported in the QSDM article. Since QSDM is only presented in a 3D setting, we restrict the empirical comparison to our 3D benchmarks. As shown in the table below, our method achieves consistently better 3D reassembly performance than QSDM under this matched comparison protocol:
>
> |Breaking Bad (Everyday)|RMSE(R)↓|RMSE(T)↓|PA↑|CD↓|
> |-|-|-|-|-|
> |QSDM|7.28|1.54|93.28|0.43|
> |Ours|5.80|1.12|95.91|0.19|
>
> |Breaking Bad (Artifact)|RMSE(R)↓|RMSE(T)↓|PA↑|CD↓|
> |-|-|-|-|-|
> |QSDM|7.80|1.76|93.01|0.97|
> |Ours|5.36|1.01|95.72|0.33|
> We will include this discussion and comparison in the revised version.
>
> # Q2: Runtime overhead relative to GARF:
>
> The additional runtime mainly arises from the quotient projection operator. However, this overhead is very small. Analytically, the extra cost consists of an $O(N)$ accumulation over fragments and then solving a $1\times 1$ (2D) or $3\times 3$ (3D) linear system, so the encoder / interaction backbone remains the dominant cost during both training and sampling.
> |Model|Train step one epoch(min)|Sampling (s) (Batch size = 16)|
> |-|-|-|
> |GARF baseline|$2.14\pm0.05$|$3.12\pm0.07$|
> |Ours|$2.15\pm0.04$|$3.17\pm0.06$|
> So the practical overhead is indeed negligible.
>
> # Q3. Sensitivity to the metric in Eq. (2):
>
> We believe the reviewer refers here to Weakness 1 (metric choice). We answer accordingly.
> Consider the weighted metric $\lVert \dot x \rVert_{M_{a,b}}^2 = 2\sum_i \left(a\lVert \dot p_i \rVert^2 + (1-a)\lVert \omega_i \rVert^2\right), 0 \leq a \leq 1$. In practice, the choice $a=.5$ does not mean that translation and rotation are treated as physically identical. Their relative influence is further modulated by the quotient projection, since this projection depends on the current fragment configuration through the inertia matrix.
> In that sense, the method already includes a data-dependent normalization mechanism, but it operates at the level of the projection rather than through a separately chosen length scale.
> In other words, the balance is not determined only by the metric coefficients.
> By contrast, choosing $a\neq .5$ would inject an additional fixed preference into that projection.
> Empirically, we found the neutral choice $a=.5$ to perform best over four settings. We report this sensitivity analysis below.
> |a/(1-a)|SVJP overlap↑|RePAIR overlap↑|B. Bad (Everyday) PA↑|FRACTURA PA↑|
> |-|-|-|-|-|
> |0.50|.91|.37|95.34|84.72|
> |0.75 |.93|.39|95.43|85.01|
> |1.00 (default)|.93|.41|95.91|85.13|
> |1.25|.92|.37|95.89|84.89|
> |1.50|.92|.35|93.62|83.82|

---

> > ### Author Rebuttal · Reviewer_MFqn · 2026-04-01
> >
> > Thank you for your answer. My problem has been solved and I will enhance my confidence.

---

### Official Review · Reviewer_KDw8 · 2026-03-10

**Soundness:** 3
**Presentation:** 3
**Significance:** 3
**Originality:** 2
**Overall Recommendation:** 4
**Confidence:** 3

**Summary:**

The paper proposes a generative model for matching/reassambling fractured object. It follows an established framework (which is not the main contribution here, but I summarize it to makes sure my understanding matches): The method learns the paths for fragments (whose geometry is encoded in a suitable way) along rigid motion to reassemble as a complete shape through a generative model (in this case a flow-matching approach, but other diffusion-style methods have been used in the past, too).

The specific problem in this paper is to make the problem frame invariant: Global rigid transformations applied to the result should not affect the outcome. This is achieved in two main steps: First, the individual parts are pre-aligned by a simple PCA-analysis with deterministic tie-breaking rules. Second, the flow field that assembles the object is setup in the tangent space of $SO(3)^n$ (for the $n$ fragments), i.e., each individual motion is represented by a linear vector in 6D space (the Lie-Algebra). Correspondingly, it is easy (and this is probably the main, neat idea in this paper) to "project out" any global rigid transformation that affects all pieces the same way.

These two measures together lead to a notable improvement in accuracy in benchmarks when comparing against prior work. The paper takes great care to make the results comparable by using comparable parts where needed, so the improvement seems to be indeed causal.

**Compliance With Llm Reviewing Policy:**

Affirmed.

**Final Justification:**

The rebuttal and the other reviews have not changed my view. I think that this is nice work, and I would be happy to see it accepted (I would recommend doing so). In my personal view, it is overall a bit incremental and specific, so while I am giving a positive recommendation it is probably not a "must have" paper for ICML.

**Key Questions For Authors:**

I do not have specific questions - the paper was very clear. Of course, please correct me if I got something wrong in my review.

**Limitations:**

I have not seen a discussion of social impact but I find it hard to imagine anything that is non-obvious and specific to this approach.
On pure technical grounds, one could discuss limitations of the generative reassembly approach as such (e.g., transferability across data sets), but this is not the main contribution here (which is frame invariance).

**Strengths And Weaknesses:**

**Strengths:**
- (Significance) The paper yields very good results, clearly outperforming prior work. The idea is simple but effective and should be easy to reproduce and/or adapt.
- (Soundness) The empirical evaluation seems to be solid, with great care taken to avoid comparing apples and oranges. I should say that am not an expert in this subfield, but it looks convincing from my (slightly outsider) point of view.
- (Presentation) The paper is written in a formal and abstract mathematical view that positions the to some extend mundane approach in a broader conceptual context of Gauge symmetry and differential geometry. This provides some extra value in inspiring related work using similar structural ideas, but it also comes with disadvantages (see below).

**Mixed:**
- (Originality) The Lie-Group approach to modeling rigid motion and dealing with invariance is not conceptually novel as such in geometry processing, but it arguably involves some a bit more advanced structural insights into the problem. The key idea (in my perception) of factoring out global rigid mappings by linear projections in the tangent space is really nice; this makes a seemingly a bit more difficult problem easy.

**Weaknesses:**
- (Significance) The scope of the contribution is limited to improving global frame invariance. While this seems to have a strong effect (which is a key strength of the submission), the broader impact to machine learning is still moderate.
- (Presentation) The very technical writing could probably easily obscure the (not so complicated) ideas to readers with a less formal background. This is a weakness, although the formal positioning has also positive effects (see above). I would recommend adding a short summary in more basic "computer graphics 101" language to attract a broader audience.
- (Soundness) A minor concern is that symmetry is broken by PCA-analysis ("global gauge fixing"), which has known short-comings in corner cases. The paper does make a good effort to justify the choices in an ablation/alternative study in the appendix, but the solution is not very elegant and might rely on data set characteristics.

**Not rated:**
- (Originality) I am not actively working in this subfield; therefore my ability to judge originality over very recent prior publications is limited. (I did not find any issues though.)

**Overall recommendation:**
I think that this is a nice piece of work and I would recommend accepting it. I am not giving a very strong recommendation because of somewhat limited novelty/originality and limited scope of impact (refining a specific reassembly technique), but if I am not missing something important (in terms of evaluation, prior work and/or soundness), my recommendation would be clearly positive.

---

> ### Author Rebuttal · Authors · 2026-03-31
>
> We thank the reviewer for the careful reading, the accurate summary, and the positive recommendation. We are especially glad that the key idea of projecting out the global rigid drift in tangent space came across clearly. In the following, we briefly clarify the three points mentioned in the weaknesses.
>
> # Significance:
> We agree that Lie groups, rigid-motion geometry, and quotient ideas are classical when taken individually. Our novelty claim is therefore not that these ingredients are new
> in isolation. Rather, the key point is that fragment reassembly requires resolving two ambiguity problems simultaneously:
> - the endpoint ambiguity, since many configurations represent the same assembled object up to a global rigid motion, and
> - the velocity ambiguity, since many tangent targets differ only by a shared global rigid drift.
>
> Our contribution is to show that these two ambiguities can be handled in a simple and unified way within a generative reassembly framework. The ingredients mentioned above are the tools that make this resolution possible in practice, rather than the deeper intuition that motivated our work. We have written the methodological section in a way that allows readers to understand how our method works and to reproduce it. As requested by reviewer EoKq, we will add a result that justifies our solution from a theoretical standpoint, showing that it has the same properties as the classical flow-matching methodology.
>
> # Presentation:
> We appreciate the suggestion and in the revised version, we will add the plain-language paragraph given in the opening statement to explain the method in more intuitive terms before introducing the gauge-theoretic formalism.
>
> # Soundness:
> We agree that PCA-based canonicalization can be ambiguous in degenerate or highly symmetric cases. However, this reflects an intrinsic ambiguity of the reconstruction problem rather than a pathology specific to our method. We address such cases with deterministic tie-breaking rules and did not observe empirical instabilities in practice (see Sec. A.2 of the supplementary material). More importantly, the PCA representation works better than other choices, and our approach does not rely solely on gauge fixing: the quotient-invariant loss remains meaningful even without gauge projection, and we have conducted an additional ablation using the quotient loss only. For more details on this point, including the new ablation and the discussion of degenerate cases, please see our response to Reviewer BtUh.

---

> > ### Author Rebuttal · Reviewer_KDw8 · 2026-04-01
> >
> > Dear authors,
> >
> > thanks for the thoughtful reply. Reading the rebuttal and the other reviews (also the discussion with Rev. BtUh) confirmed to me that there was no substantial difference in our understanding.
> > Just to clarify a very minor point: I was a bit disappointed by the canonicalization approach (as were some others, too), because it is known to be hard to find a reliable canonical pose (and yes, higher-order schemes beyond PCA might not be worth the effort, I agree). In some situations different from your scenario, this can be a big problem (in particular, in many traditional settings before DL); however, I agree that in your work, the solution is probably good enough. The canonicalization only helps to make learning easier; it does not solve the problem as such alone. Thus, I would expect to see no difference over other schemes in statistical benchmarks, and only little impact in one of the few actually ambiguous cases with 2nd order symmetric tiles. When I said "PCA is not very elegant" in my review, I really meant just that; one could in principle avoid from canonical poses and try to directly match the pieces; however, this would not fit easily with the presented pipeline, and impact is probably limited anyways. So my point was that the second aspect of your method (factoring out common transformations) is really the most interesting aspect in my perception. I would not argue against the submission because it uses PCA.
> >
> > Overall, I remain positive and I would recommend accepting the paper. I am not giving a very strong score because I think that the result is still a bit incremental (this might be a matter of personal taste/preference in the end), but I would be happy to see it accepted; it is solid work.

---

### Official Review · Reviewer_BtUh · 2026-03-13

**Soundness:** 3
**Presentation:** 3
**Significance:** 3
**Originality:** 3
**Overall Recommendation:** 4
**Confidence:** 2

**Summary:**

This paper tackles the task of reassembling $N$ fragments in $n$-dimensional space, which is a shape reconstruction task that is invariant to global rigid motions. To address the issue where training directly on $\mathcal{M}=SE(n)^{N}$ can be ill-posed, the paper proposes a geometric framework that enforces invariance by construction. Specifically, the method introduces a Global Gauge Fixing (GGF) strategy that deterministically aligns configurations, alongside a quotient-invariant Flow Matching objective that operates via orthogonal projection onto the horizontal tangent bundle. Experiments show that this unified framework improves accuracy on polygonal jigsaw puzzles and 3D fracture reassembly benchmarks.

**Compliance With Llm Reviewing Policy:**

Affirmed.

**Final Justification:**

The authors have resolved my questions. Therefore, I will maintain my positive rating.

**Key Questions For Authors:**

See the Weaknesses section above.

**Limitations:**

Yes.

**Strengths And Weaknesses:**

**Strengths**

**(1) Strong mathematical formulation.**

The proposed framework introduces a principled approach using Global Gauge Fixing and quotient-invariant Flow Matching, enforcing SE(n) invariance by construction rather than relying on ad-hoc anchoring.

**(2) Comprehensive empirical evaluation.**

The paper evaluates the method across a wide variety of 2D and 3D datasets, demonstrating its versatility and consistent improvements over state-of-the-art baselines.

**(3) Broad applicability.**

The framework provides closed-form 2D and 3D instantiations and is designed to be easily plugged into existing neural network architectures without requiring modifications to their underlying feature extractors.

**Weaknesses**

**(1) Questionable robustness of the gauge fixing strategy.**

The method relies on PCA-based gauge fixing to align the point cloud's principal axes. While the authors propose deterministic tie-breaking for degenerate cases, it remains unclear how robust this step is against highly symmetric objects.

**(2) Marginal performance gains in some 3D settings.**

While the proposed method demonstrates clear dominance in the 2D experiments, the performance improvements on certain 3D benchmarks are relatively incremental. It would strengthen the paper if the authors provided further explanation for this discrepancy.

**(3) Lack of architectural details in the main text.**

For the 2D polygon benchmarks, the authors note that they propose a new task-specific shape encoder to represent individual fragments, but all architectural details are deferred to the appendix. This makes it difficult to fully assess whether the strong 2D performance gains stem purely from the proposed geometric flow matching or partially from the new backbone design.

---

> ### Author Rebuttal · Authors · 2026-03-31
>
> We thank the reviewer for the positive assessment and for the three constructive concerns. We address them below.
> # Q1: Robustness of the gauge-fixing strategy:
> We agree that PCA-based canonicalization can be ambiguous when eigenvalues are repeated or nearly repeated. However, this is not a pathology introduced by our method, but it reflects an intrinsic ambiguity of the reconstruction problem itself: whenever the observed configuration has symmetries or near-symmetries, any canonical representation must face the same identifiability issue.
> Our deterministic tie-breaking strategy is meant to resolve such cases consistently, not to remove the underlying ambiguity.
>
> Importantly, we did not observe training instabilities in these cases. As reported in Section A.2 of the supplementary materials (“Empirical analysis on geometric degeneracies”), degenerate cases had little practical impact on optimization or reconstruction quality. In practice, they mostly arise in simple settings, e.g., with a very small number of pieces (such as N = 5 or less) or in highly symmetric synthetic examples, where the reconstruction signal remains strong enough. Moreover, Table 8 in Section A.2 shows that performance does not degrade in these cases, suggesting that, in our benchmarks, such degeneracies are mostly confined to relatively simple instances. The Bottle example in Figure 2 illustrates such a nearly degenerate case, where several fragment barycenters are close to being collinear.
>
> To further justify the use of the PCA representation Section A.2 includes a comparison to another gauge fixing method (Frechet mean) and shows that PCA work best.
> Finally, our method does not rely only on gauge fixing. The quotient-invariant loss is smooth and provides an invariant target even without gauge projection. To make this clearer, we have conducted an ablation with quotient loss only (no gauge fixing). These results (see below) show that the quotient-invariant loss alone already improves substantially over the base model, while the addition of the PCA-based strategy yields further gains.
> |SVJP| Overlap↑ | Precision↑ | Recall↑ |
> |-|-|-|-|
> | Base + Quotient loss (no gauge fixing)|.92|.89|.89|
>
> |Breaking Bad (Everyday)|RMSE(R)↓|RMSE(T)↓|PA↑|CD↓|
> |-|-|-|-|-|
> |Base backbone (no invariant component)|9.67|3.23|91.48|0.76|
> |+ Gauge fixing only|6.98|1.54|94.76|0.26|
> |+ Quotient loss only|6.07|1.29|95.34|0.22|
> |+ Gauge fixing + quotient loss|5.74|1.13|95.54|0.19|
> |+ Gauge fixing + quotient loss + test-time projection|5.80|1.12|95.91|0.19|
>
> |Breaking Bad (Artifact)|RMSE(R)↓|RMSE(T)↓|PA↑|CD↓|
> |-|-|-|-|-|
> |Base backbone (no invariant component)|10.02|3.98|90.21|0.98|
> |+ Gauge fixing only|6.02|1.36|94.93|0.41|
> |+ Quotient loss only|5.76|1.30|95.01|0.37|
> |+ Gauge fixing + quotient loss|5.41|1.12|95.23|0.34|
> |+ Gauge fixing + quotient loss + test-time projection|5.36|1.01|95.72|0.33|
>
> # Q2: Why are the gains more modest in 3D?
> The more modest gains in 3D mainly stem from the fact that 3D object reassembly already benefits from highly informative fracture features.
> As shown by GARF and related works, pretrained fracture encoders significantly reduce the ambiguity of the task: smooth outer surfaces contribute little, while fracture interfaces are highly distinctive and strongly constrain the matching. As a result, prior methods are already very strong in 3D.
>
> In contrast, 2D boundaries are much less informative. They are only 1D and in most 2D datasets, nearly all fragment boundaries are candidate matching regions, so the search space remains much larger. In that setting, the room for improvement is larger, and the benefit of removing the global gauge ambiguity with our method becomes more visible. We will clarify this point in the final version.
> # Q3: Role of the backbone in the 2D results:
> First, results on other 2D baseline such as RePAIR, our results are obtained using the same backbone as prior SOTA, so the improvement comes only from our invariant framework. On the other hand, on the SVFP dataset, the setting is different because Hosseini et al.'s PuzzleFusion architecture has important limitations as it only applies to polygons with a low number of vertices. Still, to isolate the effect of our method more clearly, we also tested our quotient-invariant loss on top of the PuzzleFusion architecture. The performance still improves substantially as shown in the table below.
> This confirms that a large part of the improvement indeed comes from the invariant formulation rather than from the new encoder alone.
>
> |Method|Overlap↑ (0%)|Overlap↑ (.5%)|Overlap↑ (1%)|Precision↑ (0%)|Precision↑ (.5%)|Precision↑ (1%)|Recall↑ (0%)|Recall↑ (.5%)|Recall↑ (1%)|
> |-|-|-|-|-|-|-|-|-|-|
> |Hosseini et al., 2023|.76|.71|.60|.80|.72|.60|.74|.64|.44|
> |Ours with Hosseini et al. architecture|.89|.85|.70|.89|.84|.67|.86|.80|.69|
>
> We hope this clarifies the reviewer’s concerns, and we will incorporate these additional ablations and explanations in the revised version.

---

> > ### Author Rebuttal · Reviewer_BtUh · 2026-04-04
> >
> > I appreciate the authors’ rebuttal. All my questions were resolved, particularly regarding the robustness of the gauge-fixing strategy.

---

### Decision · Program_Chairs · 2026-04-30

**Decision:**

Accept (regular)

**Comment:**

This paper proposes an SE(n)-invariant framework for flow matching on multi-body rigid assembly problems, using PCA-based gauge fixing and a closed-form horizontal projection to factor out global rigid-body drift. All four reviewers recommend acceptance (scores: 5, 5, 4, 4) with all concerns fully resolved after rebuttal.

Strengths:
- Principled geometric framework that enforces SE(n) invariance by construction rather than by ad-hoc anchoring or augmentation
- Clean closed-form horizontal projection for SE(2) and SE(3), differentiable and efficient (O(N) accumulation + small linear solve)
- Proven theoretical guarantee that the quotient-invariant loss preserves the CFM correct-gradient property
- Comprehensive evaluation across 6 benchmarks (2D jigsaw, floorplans, archaeological reconstruction, 3D fracture assembly) with consistent improvements
- Thorough ablation isolating each component's contribution in both 2D and 3D (provided in rebuttal)
- Plug-in design: works on top of existing backbones without architectural modification
- Strong rebuttal: added 3D ablation, QSDM comparison, missing/extra piece evaluation, metric sensitivity analysis, statistical significance, and theoretical guarantee

Weaknesses (minor):
- The individual ingredients (PCA alignment, Lie algebra velocity representation, inertia-momentum projection) are classical in shape analysis and mechanics. The novelty lies in their assembly for generative flow matching, not in the components themselves.
- 3D improvements over GARF are modest, as GARF's fracture-aware features already reduce ambiguity significantly
- PCA gauge fixing can be discontinuous when the covariance matrix has near-repeated eigenvalues (e.g., highly symmetric configurations). All four reviewers raised this concern. The authors demonstrated it is empirically benign and that the quotient loss alone (without gauge fixing) already provides most of the benefit, so the method is not dependent on PCA stability.

Recommendation: Accept. Unanimous positive consensus with thorough rebuttal. The framework is mathematically sound, practically useful as a plug-in, and well-validated.

Revision requests for camera-ready:
- Add the plain-language summary of the method before the formal development (as suggested by Reviewer KDw8 and agreed by authors)
- Include the 3D ablation table and QSDM comparison from the rebuttal
- Include the metric sensitivity analysis and statistical significance results
- Fix the missing citation in App. C.3 and other typos noted by Reviewer EoKq
- Fix several reference errors: (a) Jiang et al., 2023 "SE(3)-DiffusionFields" has wrong authors (should be Urain et al.) and wrong venue (ICRA 2023, not NeurIPS); (b) Watson et al., 2023 has wrong title (cited title belongs to Anishchenko et al. 2021; correct title is "De novo design of protein structure and function with RFdiffusion"); (c) Paikin & Tal, 2015 has wrong title; (d) Zhang et al., 2014 does not exist (hallucinated?)